



# Observations and simulations of the seasonal evolution of snowpack cold content and its relation to snowmelt and the snowpack energy budget

Keith S. Jennings[1,2], Timothy G.F. Kittel[2], and Noah P. Molotch[1,2,3]

5  [1]Geography Department, University of Colorado Boulder, 260 UCB, Boulder, Colorado 80309, USA
[2]Institute of Arctic and Alpine Research, University of Colorado Boulder, 450 UCB, Boulder, CO 80309, USA
[3]NASA Jet Propulsion Laboratory, 4800 Oak Grove Drive, Pasadena, CA 91109, USA

*Correspondence to*: Keith S. Jennings (keith.jennings@colorado.edu)

**Abstract.** Cold content is a measure of a snowpack's energy deficit and is a linear function of snowpack mass and
10  temperature. Positive energy fluxes into a snowpack must first satisfy the remaining energy deficit before snowmelt runoff
begins, making cold content a key component of the snowpack energy budget. Nevertheless, uncertainty surrounds cold
content development and its relationship to snowmelt, likely because of a lack of direct observations. This work clarifies the
controls exerted by air temperature, precipitation, and negative energy fluxes on cold content development and quantifies the
relationship between cold content and snowmelt timing and rate at daily to seasonal time scales. The analysis presented
15  herein leverages a unique long-term snow pit record along with validated output from the SNOWPACK model forced with
23 water years (1991–2013) of quality controlled, infilled hourly meteorological data from an alpine and subalpine site in the
Colorado Rocky Mountains. The results indicated that precipitation exerted the primary control on cold content development
with snowfall responsible for 84.4% and 73.0% of simulated gains in the alpine and subalpine, respectively. A negative
surface energy balance—primarily driven by sublimation and longwave radiation emission from the snowpack—during dry
20  periods provided a secondary pathway for cold content development, and was responsible for the remaining 15.6% and
27.0% of cold content additions. Non-zero cold content values were associated with reduced snowmelt rates and delayed
snowmelt onset at daily to sub-seasonal time scales. These results suggest that the information provided by cold content
observations and/or simulations is most relevant to snowmelt processes at shorter time scales, and may help water resource
managers to better predict melt onset and rate.

25  **1 Introduction**

Cold content is a key component of the snowpack energy budget as it represents the internal energy deficit that must
be overcome before snowmelt runoff can begin. It is a linear function of snowpack temperature and snow water equivalent
(SWE), whereby colder snowpacks with greater SWE have increased energy deficits. Until cold content is satisfied, positive
energy fluxes go towards raising the internal snowpack temperature to an isothermal 0°C and any surface melt that is





produced may be refrozen in the colder lower layers of the snowpack. In this regard, cold content influences snowmelt timing and rate, which are of critical importance to various ecohydrologic and cryospheric processes, including: streamflow generation (Barnhart et al., 2016; Regonda et al., 2005), water resources availability (Barnett et al., 2005; Christensen et al., 2004; Mankin et al., 2015; Stewart, 2009), water uptake by vegetation (Winchell et al., 2016), soil moisture (Harpold and

Molotch, 2015), flooding (Jennings and Jones, 2015; Kampf and Lefsky, 2016), and land surface albedo (Déry and Brown, 2007), among others.

Cold content can be estimated using at least one of three primary methods: 1) As an empirical function of air temperature (e.g., Anderson, 1976; Seligman et al., 2014; United States Army Corps of Engineers, 1956); 2) As a function of precipitation and air temperature (e.g., Cherkauer et al., 2003; Lehning et al., 2002b; Wigmosta et al., 1994) or wet bulb

temperature (Anderson, 1968) during precipitation; and 3) As a residual of the snowpack energy balance (e,g., Andreadis et al., 2009; Cline, 1997; Lehning et al., 2002b; Marks and Winstral, 2001). In general, simple temperature-index models employ method 1, while both 2 and 3 are utilized in physics-based snow models. These methods suggest that cold content develops through both meteorological and energy balance processes, but few direct comparisons to observed cold content exist. This is likely due to the inherent difficulty in measuring cold content, which requires either time-intensive snow pits or

co-located snow depth, density, and temperature measurements (Burns et al., 2014; Helgason and Pomeroy, 2011; Marks et al., 1992; Molotch et al., 2016). The lack of validation data introduces significant uncertainty into the dominant process by which cold content develops. Thus, it is not known whether cold content is primarily a function of air temperature (method 1), snowfall (method 2), or a negative surface energy balance (method 3).

Early work from California's Sierra Nevada mountains indicated cold content developed in the snowpack mainly

through a negative surface energy balance. The authors reported the monthly net flux (i.e., change in cold content) ranged from -34 to -61 W m$^{-2}$ from November through April at an exposed site and -8 to -66 W m$^{-2}$ from November through February at a sheltered site (Marks and Dozier, 1992). However, such negative fluxes would result in physically unrealistic monthly internal snowpack temperature changes. Even persistent slightly negative flux values, as reported elsewhere in the literature (Armstrong and Brun, 2008), would result in implausibly low snowpack temperatures. It can be inferred that any

process producing anomalously low snowpack temperatures either misidentifies or overestimates the importance of a particular meteorological or energy balance mechanism.

Furthermore, the degree to which cold content controls snowmelt timing and rate at daily to seasonal timescales is relatively uncertain. Work from the southwestern United States suggests increased cold content may delay seasonal melt timing (Molotch et al., 2009) and the inclusion of cold content generally improves meltwater outflow predictions in point

and distributed snowmelt models of varying degrees of physical complexity (Bengtsson, 1982; Jepsen et al., 2012; Livneh et al., 2010; Mosier et al., 2016; Obled and Rosse, 1977). However, two empirical studies indicated the energy required to satisfy cold content may be relatively small in comparison to the energy required to saturate an already isothermal snowpack (Bengtsson, 1982; Seligman et al., 2014).





Given the above uncertainties, we aim to improve understanding of the processes controlling cold content development and the relationship between cold content and snowmelt timing and rate by utilizing observations from a long-term snow pit record and forcing a physics-based snow model with a quality controlled, serially complete meteorological dataset. Analyses performed on the observations and simulation data are focused on answering the following research questions:

1. What are the meteorological and energy balance controls on cold content development?
2. How does cold content affect snowmelt timing and rate on seasonal, sub-seasonal, and daily time scales?

## 2 Study site and snow pit and forcing data

The Niwot Ridge Long Term Ecological Research site (LTER) is located on the eastern slope of the Continental Divide in the Rocky Mountains of Colorado, USA (Fig. 1). The entirety of the LTER is situated above 3000 m with treeline occurring at approximately 3400 m (Williams et al., 1998). Dominant vegetation in the subalpine is lodgepole pine, aspen, Engelmann spruce, subalpine fir, and limber pine (Burns et al., 2014). The alpine is characterized by several tundra vegetation communities of grasses, forbs, and shrubs, whose distribution is linked to patterns of snow depth and soil moisture (Walker et al., 1993, 1994).

There are multiple meteorological stations within the boundaries of the Niwot Ridge LTER, but this work focuses on the two sites with long-term snow pit records: alpine (3528 m) and subalpine (3022 m), named Saddle and C1, respectively (Fig. 1). We employed an additional high alpine station (D1, 3739 m) in the meteorological data infilling procedure (Appendix A), but did not perform model simulations there due to a lack of snow pit validation data. From 2008 to 2012, annual precipitation in the alpine and subalpine averaged 1071 mm and 752 mm, respectively (Knowles et al., 2015) and the ratio between above- and below-treeline precipitation varies annually as a function of upper-air flow regimes (Kittel et al., 2015). The majority of annual precipitation is snow, with estimates of the proportion of snowfall ranging from 63% to 80% of total precipitation (Caine, 1996; Knowles et al., 2015). Dominant wind direction is westerly, but the subalpine site also experiences easterly flow during intermittent upslope events (Blanken et al., 2009; Burns et al., 2014). Elevated wind speeds in the alpine, averaging 10-13 m s$^{-1}$ in winter, exert a primary control on patterns of snow erosion and deposition with snow depth being highly variable as a result (Erickson et al., 2005; Jepsen et al., 2012; Litaor et al., 2008). Snow depths in the alpine can range from 0 m over wind-scoured tundra to upwards of 5 m in drifts on the lee side of terrain features or in gullies. Additionally, blowing snow occurs frequently during winter months in the alpine due to high winds, reaching a maximum in January (Berg, 1986).

Regular snow pit measurements began in 1995 in the alpine and 2007 in the subalpine, and were taken at weekly to monthly intervals from the middle of January through the end of May in most snow seasons (Williams, 2016). A total of 292 alpine and 147 subalpine snow pit records were used in this study (Table S1). The alpine snow pit represents conditions typical of the above-treeline snowpack as it is not in an area of pronounced snow erosion or deposition. The subalpine snow pit is





located in a small clearing of lodgepole pine, typical of vegetation conditions in the below-treeline areas. Measurement protocol follows Williams et al. (1999): Snow density is measured for each 10 cm layer using a wedge-shaped 1 L density cutter (10 cm × 10 cm × 20 cm) and snow temperature is recorded every 10 cm with dial-stem thermometers. Snow pit measurements enable per-layer and depth-weighted calculations of SWE and cold content:

$$SWE = \frac{\rho_s}{\rho_w}d_s \tag{1}$$

$$CC = c_i\rho_s d_s(T_s - T_m) \tag{2}$$

where $\rho_s$ and $\rho_w$ are the density of snow and liquid water, respectively (kg m$^{-3}$), $d_s$ is snow depth (m), $CC$ is cold content (MJ m$^{-2}$), $c_i$ is the specific heat of ice (2.1 × 10$^{-3}$ MJ kg$^{-1}$ °C$^{-1}$), $T_s$ is the snow temperature (°C), and $T_m$ is the melting temperature of snow (0°C). Snow pit analyses focused on water years (WY, 1 October from the previous calendar year through 30 September) 2007 through 2013, the period for which overlapping snow pit data were available. The full period of record in the alpine (WY1995–WY2013) was used for model validation.

Hourly meteorological data have been collected at the LTER since 1990, but the record suffers from quality control issues and periods of missing data. Recent research has shown the quality of snow model output depends on having accurate forcing data (e.g., Förster et al., 2014; Lapo et al., 2015; Raleigh et al., 2015, 2016; Schmucki et al., 2014). Measurements were therefore subjected to an extensive quality control and infilling protocol (Appendix A) to produce a serially complete, hourly dataset with observations of air temperature, relative humidity, incoming solar radiation, wind speed, and
precipitation, plus an estimate of downwelling longwave radiation based on air temperature, relative humidity, and incoming solar radiation.

**3 Methodology**

Observations from the Niwot Ridge LTER snow pit record and validated output data from physics-based snow model simulations were employed to answer the two research questions proposed in Sect. 1. We assessed the meteorological
controls on cold content development using measurements of cumulative precipitation and the cumulative mean of air temperature for the full period of record at both sites. We focused the analysis on snow pit observations and simulations between 1 December and the date of peak cold content, the main period of cold content development. We then quantified the contribution of the snowpack energy budget to cold content development using the change in internal energy between pit observations as well as the energy flux estimates provided by the snow model simulations. Model output was also used to
assess the effect of peak cold content magnitude and timing on snowmelt rate and timing at seasonal and daily time scales. Additionally, we note that in this paper an "increase" in cold content refers to the value increasing in magnitude and becoming more negative (i.e., the energy deficit is becoming greater). A "decrease" of cold content occurs when the value becomes less negative and approaches 0 MJ m$^{-2}$.





### 3.1 Snow pit analysis

Mean characteristics of and differences between the alpine and subalpine snow pits were quantified using data from WY2007–WY2013, the seven years for which there were overlapping observations. To assess the control each meteorological quantity exerted on cold content, we used the cumulative mean of air temperature and cumulative

precipitation as the independent variables with observed cold content acting as the dependent variable in ordinary least squares regression. The strength of the relationship was quantified using the coefficient of determination, $r^2$, while the p-value of the regression slope indicated statistical significance. Additionally, in order to evaluate whether large persistent negative energy balances were consistent with patterns of cold content development, we calculated the net energy flux between snow pit observations:

$$Q_{net} = \frac{\Delta CC}{(86,400 \times \Delta t)} \qquad (3)$$

where $Q_{net}$ is the net flux (W m$^{-2}$), $\Delta CC$ is the change in cold content (J m$^{-2}$), 86,400 is the conversion factor between days and seconds, and $\Delta t$ is the number of days between snow pit observations. Snow pit cold content in this context integrates the effects of incoming and outgoing fluxes by providing a measure of the change in the internal energy of the snowpack independent of any flux measurements or estimations.

### 3.2 Snow model simulations

### 3.2.1 Model description

In order to improve on the temporal resolution of the snow pit observations, expand the study period, and quantify components of the energy budget, we employed the complex, physics-based, multi-layer, one-dimensional SNOWPACK model (Bartelt and Lehning, 2002; Lehning et al., 2002a, 2002b). This model was selected because previous studies have shown complex, multi-layer models more accurately partition the snowpack energy budget and better represent internal

processes (Blöschl and Kirnbauer, 1991; Boone and Etchevers, 2001; Essery et al., 2013; Etchevers et al., 2004). It has also been utilized previously to simulate the snowpack energy budget at the Niwot Ridge LTER (Meromy et al., 2015) and has been validated in the Rocky Mountains of Montana (Lundy et al., 2001). SNOWPACK is forced with air temperature, relative humidity, wind speed, incoming solar radiation, incoming longwave radiation, and precipitation at an hourly or higher temporal resolution. The model discretizes the snowpack into a variable number of finite elements that change with

the addition of new snow, mass loss through snowmelt and sublimation, and densification via compaction. Each layer is composed of water in liquid, solid, and gas phases, all of which are assumed to have the same temperature. The numerical model in SNOWPACK is governed by four differential equations that account for the conservation of energy, mass, and momentum. Explicit routines are included for heat transfer, water transport, vapor diffusion, and phase changes. In addition, the model features quasi-physical estimations of snow microstructure and snow grain metamorphism. These properties, in

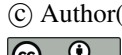



turn, control the rate of heat conduction and settling within the snowpack. SNOWPACK also models the penetration of shortwave radiation and wind pumping in the upper layers of the snowpack.

The bulk Richardson number stability correction was used for computing turbulent fluxes in both the alpine and subalpine. Although Monin-Obukhov similarity theory options are available, this stability correction generally performed

worse relative to the bulk Richardson number in our preliminary simulations as well as in the work of others (Essery et al., 2013). Additionally, the two-layer SNOWPACK canopy module (Gouttevin et al., 2015) was activated for the subalpine simulations. Parameters for the canopy module were calibrated using a series of 100 Monte Carlo simulations with value ranges bounded by representative estimates of leaf area index, vegetation height, direct canopy throughfall, and wind speed reduction. Model SWE output in the subalpine proved most sensitive to the wind speed reduction parameter, likely due to the

siting of the anemometer as noted in Appendix A. Using un-corrected observed wind speed as a model input led to a physically unrealistic amount of snow sublimation.

### 3.2.2 Model simulations, validation, and analysis

SNOWPACK simulations were performed in the alpine and subalpine for WY1991–WY2013 and forced with the quality controlled, infilled hourly meteorological data detailed in Appendix A. This time range included the lowest (WY2002: 178

mm) and second highest (WY1996: 523 mm) peak SWE observations in the period of record (WY1981–WY2017) at the Niwot Snowpack Telemetry (SNOTEL) station (3020 m), which is located within the Niwot Ridge LTER boundary, less than 1 km from the subalpine snow pit and meteorological tower. Thus, the analysis covered a wide range of feasible snowpack conditions, from pronounced snow drought to peak SWE values greater than 150% of average, according to the SNOTEL observations.

To ensure the simulations were suitable for in-depth analyses, we validated model SWE, snowpack temperature, and cold content values on the snow pit observations. We pursued this multi-validation approach because our work focuses on the internal energy of the snowpack and recent research has shown the output from snow model simulations is more reliable when several variables are used in model evaluation (Lapo et al., 2015). Modeled subalpine SWE estimates were also evaluated using observed SWE at the Niwot SNOTEL site. For each quantity of interest, we assessed model performance

using the coefficient of determination and mean bias. To improve model output, we corrected precipitation measurements relative to snow pit and SNOTEL SWE observations (Appendix A) and optimized the canopy parameters for subalpine simulations (Sect. 3.2.1).

We then used the validated output from SNOWPACK to quantify the controls on cold content development and snowmelt processes at a finer temporal resolution than the weekly to monthly snow pit observations. To evaluate the

meteorological processes controlling cold content development, we used the same methods employed in the snow pit observations outlined above (Sect. 3.1). Additionally, we quantified the contributions of the simulated snowpack energy balance to cold content development:





$$Q_{net} = Q_{SW} + Q_{LW} + Q_H + Q_{LE} + Q_G + Q_R - Q_M \qquad (4)$$

where $Q_{SW}$ is net shortwave radiation, $Q_{LW}$ is net longwave radiation, $Q_H$ is sensible heat flux, $Q_{LE}$ is latent heat flux, $Q_G$ is ground heat flux, $Q_R$ is the heat advected by precipitation, and $Q_M$ is the energy available for melt (all W m⁻²).

Simulation results were also used to quantify the control cold content exerts on snowmelt timing and rate at multiple time scales. At the seasonal time scale, we set snowmelt onset to correspond to the date of peak SWE and snowmelt

rate to the ablation slope, which is the average daily snowmelt rate between the date of peak SWE and the date at which SWE first equals 0 mm (e.g., Barnhart et al., 2016; Trujillo and Molotch, 2014). At sub-seasonal time scales, we calculated snowmelt timing and rate in time windows from 1 d to 30 d, with a corresponding cold content value at day zero. Finally, we used the cold content at 6AM ($CC_{6AM}$) to evaluate the effect of cold content on snowmelt timing and rate at sub-daily time scales. For the sub-seasonal and sub-daily time scales above, we set snowmelt timing to be the first instance of simulated

snowmelt runoff and snowmelt rate to be the mean rate for the time window.

## 4 Results

### 4.1 Snow pit observations of cold content

Snow pit observations showed daily and peak annual snowpack cold content were consistently greater in the alpine than subalpine (Fig. 2). From WY2007–WY2013, mean peak cold content was 2.6 times greater in the alpine than subalpine,

while mean peak SWE was 2.1 times greater in the alpine (Table 1). Peak cold content and peak SWE respectively occurred 33 d and 10 d later in the alpine than subalpine. The temporal gap between peak cold content and peak SWE was also 23 d shorter in the alpine, indicating greater energy exchange between the snow and atmosphere at this site during the main time of snowpack ripening. Mean $Q_{net}$ for this period, as estimated using Eq. 3, was 1.2 W m⁻² and 0.4 W m⁻² in the alpine and subalpine, respectively.

From 1 December to the date of snow pit observation, increased cumulative precipitation was associated with increased cold content at both sites (Fig. 3). Cumulative precipitation explained 55% and 17% of the variance in cold content in the alpine and subalpine, respectively. The relationship was statistically significant at the 99% level at both sites despite the low coefficient of determination in the subalpine. Conversely, the cumulative mean of air temperature had no statistically significant relationship with snowpack cold content, explaining less than 1% of the variance at both sites (not shown). These

results indicate that snowfall exerts the primary control on cold content development. This is likely due to the increased variability of winter precipitation, the coefficient of variation of which is 2.9 and 2.7 times greater than that of air temperature in the alpine and subalpine, respectively.

Snow pit observations were also used to infer $Q_{net}$ by quantifying the change in cold content between two points in time (Eq. 3). During periods of SWE accumulation, $Q_{net}$ was typically near 0 W m⁻² (Fig. 4a), indicating a large negative

energy balance was not responsible for cold content development. The average flux in the alpine was greater in magnitude during this period than in the subalpine, and both distributions were left-skewed as the energy balance was typically negative



from snowfall-driven cold content increases. Changing the analysis to snow pit observations when melt occurred (Fig. 4b) led to a pronounced right-skew in the flux distribution with values again of a higher magnitude in the alpine.

## 4.2 Model SWE, snowpack temperature, and cold content validation

SNOWPACK simulations reproduced observed snow pit SWE patterns at both sites, with a higher coefficient of
determination and lower bias in the subalpine than alpine (Fig. 5a,b; Table 2). Subalpine simulations were also in line with daily SWE observations from the Niwot SNOTEL (Table 2). Simulated snowpack temperature had a slight warm bias of 1.1°C in the alpine and 0.6°C in the subalpine (Fig. 5c,d, Table 2), while cold content was overpredicted in the alpine and underpredicted in the subalpine (Fig. 5e,f, Table 2). Modeled annual peak SWE and peak cold content were also similar to the previously reported pit values for WY2007 through WY2013 (Table 2). Additionally, simulated LTER subalpine peak
cold content values were within the range of those reported in a simulation of a subalpine snowpack (-2.2 to -1.7 MJ m$^{-2}$) at the nearby Fraser Experimental Forest during NASA's Cold Land Processes Experiment (Marks et al., 2008). Direct observations of snow surface sublimation were not available for comparison, but modeled sublimation rates were in line with other values reported in the literature for alpine and subalpine areas in the Colorado Rocky Mountains (Berg, 1986; Hood et al., 1999; Knowles et al., 2012; Molotch et al., 2007; Sexstone et al., 2016). On average, SNOWPACK-simulated
sublimation represented 28.8% (383 mm) and 11.4% (53 mm) of snow-season precipitation in the alpine and subalpine, respectively.

## 4.3 Meteorological and energy balance controls on cold content development: Simulation results

### 4.3.1 Primary control: Snowfall

Similar to the snow pit observations, simulated cold content was strongly related to cumulative precipitation, indicating cold
content developed primarily through the addition of new snowfall (Fig. 6a,b). Of the two sites, cumulative precipitation explained more of the variance in the alpine where cold content increased near-monotonically with precipitation. The subalpine snowpack frequently approached an isothermal state in the winter with cold content fluctuating between gains during snowfall and losses during dry periods. Additionally, the cumulative mean of air temperature explained little of the variance in simulated cold content (Fig. 6c,d). In general, large increases in cold content were not associated with decreases
in air temperature, meaning periods of below-average air temperature did not significantly contribute to cold content development. These simulations confirm the result of the snow pit observations, namely that of the two main meteorological quantities, precipitation exerts the primary control on cold content development.

Discretizing snow season days into those with and those without snowfall further clarifies the relationship between cold content development and precipitation. Figure 7 shows the monthly differences between dry and wet days in the alpine
and subalpine in terms of cold content gains and losses. Wet days were commonly associated with cold content gains, particularly in December, January, and February when precipitation was coincident with low air temperatures. Dry days,



conversely, were associated with decreases in snowpack cold content, indicating a positive surface energy balance warmed the snowpack between snowfall events. Magnitudes were typically greater in the alpine where colder temperatures and increased precipitation led to greater cold content gains on wet days, while higher wind speeds facilitated increased rates of energy transfer and cold content losses on dry days.

### 4.3.2 Secondary control: Negative surface energy balance

Although precipitation exerted the primary control on cold content development, gains occurred during dry periods, as well. The majority of non-precipitation cold content additions took place at night (1800 h through 0600 h) when simulations showed the snowpack cooled through a negative surface energy balance, with $Q_{net}$ averaging -5.1 W m$^{-2}$ in the alpine and -6.2 W m$^{-2}$ in the subalpine (Fig. 8a,b). $Q_{LE}$ and $Q_{LW}$ were the primary negative energy balance terms at both sites, while $Q_H$ and $Q_G$ contributed the main positive energy inputs. $Q_{net}$ values were similar for daytime cooling, with both sites averaging -5.2 W m$^{-2}$. Again, the main negative energy balance contributions came from $Q_{LE}$ and $Q_{LW}$. In total, nighttime cold content additions outnumbered daytime additions by a 2.7:1 ratio in the alpine and 3.7:1 in the subalpine (Fig. 8c,d).

### 4.3.3 Comparing the relative importance of cold content development processes

Overall, precipitation exerted the primary control on cold content development relative to air temperature and a negative energy balance at both sites. The number of wet days with cold content increases exceeded the number of dry days with increases in the alpine by a 4.2:1 ratio, with wet days responsible for 438% more cold content additions than dry days. On an average annual basis in the alpine, wet days contributed -12.5 MJ m$^{-2}$ to cold content development and dry days -2.3 MJ m$^{-2}$. As previously noted, the effect of precipitation was smaller in the subalpine in terms of both the variance explained by cumulative precipitation and the ratio of wet-to-dry cold content gains. Wet days in the subalpine were responsible for 166% more cold content gain than dry days, generating -4.1 MJ m$^{-2}$ and -1.5 MJ m$^{-2}$ of cold content development on an annual basis, respectively.

Although cumulative mean air temperature had little effect on seasonal cold content development, air temperature did influence the amount of cold content added to the snowpack per snowfall event. Figure 9 shows the daily change in cold content in the alpine and subalpine relative to daily total precipitation (a,b), and cold content from precipitation (c,d). Here the cold content from precipitation was calculated as in Eq. 2 but $T_s$ was replaced with air temperature and $d_s$ was replaced by the depth of precipitation. At both sites, the cold content from precipitation explained more of the variance in daily change in cold content than daily total precipitation alone, indicating air temperature provides a secondary control on cold content development during snowfall events. Confirming previous results, the control exerted by precipitation on cold content development was stronger in the alpine than subalpine.



### 4.4 The effect of cold content on snowmelt rate and timing

On seasonal time scales, the magnitude of annual peak cold content had a delaying, but statistically non-significant effect on snowmelt onset, according to both observations and simulations (not shown). However, using the 23 y of snowpack simulations, we found the date of peak cold content and spring precipitation—defined here as the total precipitation between

the date of peak cold content and peak SWE—accurately predicted melt onset. A multiple linear regression (MLR) using peak cold content day of WY (DOWY) and spring precipitation as the predictor variables explained 84.7% and 61.4% of the variance in peak SWE DOWY in the alpine and subalpine, respectively (Fig. 10). At both sites, later peak cold content and increased spring precipitation delayed melt onset. In the alpine, the MLR predicted a 1 d delay in snowmelt timing per 1.6 d later in peak cold content timing or 8.8 mm extra spring precipitation. These values shifted to 2.3 d and 5.9 mm,

respectively, in the subalpine. Furthermore, we found cold content exerted no statistically significant control on the seasonal snowmelt rate. Rather, statistically significant increases in the ablation slope were associated with later peak SWE timing and increased peak SWE magnitude.

While peak cold content magnitude exerted little control on seasonal snowmelt timing and rate, the simulations indicated increased cold content had a damping effect on snowmelt timing and rate at sub-seasonal time scales from 1 d to

30 d. Greater initial cold content values were associated with decreased snowmelt rates (Fig. 11a,b) and longer delays between day zero and the day of first snowmelt (Fig. 11c,d). All relationships were significant at the 99% level, except for the effect of cold content on snowmelt timing for the 1 d time window in the subalpine. Simulated melt rates in the alpine only exceeded 40 mm $d^{-1}$ when initial cold content was between -0.1 MJ $m^{-2}$ and 0 MJ $m^{-2}$. The same initial cold content range was responsible for all simulated melt rates greater than 15 mm $d^{-1}$ in the subalpine. Examining only the 30 d window

for snowmelt timing revealed further patterns at the two sites. Initial cold content explained 47.3% of the variance in time to first melt in the alpine and 37.6% in the subalpine using ordinary least squares regression. An initial cold content increase of 1.0 MJ $m^{-2}$ led to a 3.7 d delay in snowmelt in the alpine and 12.1 d in the subalpine.

To examine the control of cold content on daily snowmelt rate and timing, we used $CC_{6am}$ to represent the energy state of the snowpack at time t = 0 for each day. Figure 12a,b shows melt rates did not increase until $CC_{6AM}$ neared 0 MJ $m^{-2}$

in the alpine and subalpine. Both the number of melt days and the daily melt rate were greater when $CC_{6AM}$ = 0 MJ $m^{-2}$. The proportion of daily melt occurring on days when $CC_{6AM}$ = 0 MJ $m^{-2}$ ranged from 75.0% in the alpine to 79.5% in the subalpine. Mean melt rates were also greater when there was no energy deficit to satisfy in the alpine (21.1 vs. 14.3 mm $d^{-1}$) and subalpine (9.7 vs. 6.2 mm $d^{-1}$). Additionally, non-zero $CC_{6AM}$ values were associated with delayed snowmelt onset (Fig. 12c,d). The mean time between 6AM and simulated snowmelt onset was 2.3 h in the alpine and 2.8 h in the subalpine when

$CC_{6AM}$ = 0 MJ $m^{-2}$. These values shifted to 5.7 h and 6.7 h, respectively, when $CC_{6AM} \neq 0$ MJ $m^{-2}$. Thus the presence of cold content produced a 3.4 h delay in alpine snowmelt onset and 3.9 h in the subalpine. These data indicate that even small energy deficits had a damping effect on daily snowmelt rate and timing.




## 5 Discussion

### 5.1 Representation of cold content development processes in snow models

In Sect. 1 we noted the three main methods by which cold content is represented in snow models. Temperature index models typically compute cold content as an empirical function of air temperature (method 1), while physical models estimate cold

content as a function of precipitation and the air temperature during precipitation (method 2) and/or as a residual of the snowpack energy balance (method 3). A model comparison is outside of the scope of this work, but the results presented above suggest the representation of cold content development within snow models would be most consistent with physical processes using method 2. We found air temperature had little influence on cold content development except when included as a variable in computing the cold content of new snowfall. Prior work from the subalpine site of the Niwot Ridge LTER

showed a weak relationship between cold air temperatures and snowpack cooling and that periods of snowpack cooling were generally coincident with clear skies and longwave emission from the snowpack (Burns et al., 2014). Thus, method 1 is likely to misrepresent cold content development processes and incorrectly estimate cold content magnitude in snow model simulations.

     Our results also indicate method 3 provides utility in simulating cold content development, but to a lower degree than

method 2. Additionally, we found no evidence in either the simulations or observations of consistent, large negative energy balances producing cold content. Rather, the energy balance was typically near zero before peak SWE and only became significantly positive once melt commenced. Periods with a negative surface energy balance were generally short, associated with nighttime cooling from $Q_{LE}$ and $Q_{LW}$, and small in magnitude, averaging $> -6.0$ W m$^{-2}$. Marks and Winstral (2001) similarly noted the simulated energy balance in a semi-arid mountain basin was generally near 0 W m$^{-2}$ until the melt season.

Overall, these findings imply snowpack cold content development is primarily a function of method 2 and that large flux-driven increases in cold content are unlikely, even in areas where the energy balance plays a larger relative role (e.g., the subalpine site studied here).

### 5.2 Differences between cold content development controls in the alpine and subalpine

Despite only a 506 m elevation difference between the two sites, the role of a negative energy balance in developing cold

content in the subalpine was approximately double that of the alpine. Simulations of snowpack temperature indicated the increased sensitivity was likely due to the shallower subalpine snow depth. Diurnal snowpack temperature range generally decreases with depth (e.g., Burns et al., 2014; DeWalle and Rango, 2008; Sturm et al., 1995) and our simulations showed daily fluctuations to be largest in the snowpack's upper layers, converging towards 0.3°C to 0.5°C as depth exceeded 500 mm (Fig. 13). This is the same depth at which the insulating effects of snow on soil temperature become marginal (Slater et

al., 2017). Likely this is because the penetration of incoming shortwave radiation and sensible heat transfer through windpumping are limited to the top portion of the snowpack (Albert and McGilvary, 1992; Colbeck, 1989a, 1989b; Lehning et al., 2002b), while the low thermal conductivity of snow modulates energy transfer below the active upper layers (Sturm et



al., 1997). In this case, proportionally more of the shallower subalpine snowpack was interacting with surface energy exchange, making it more sensitive to positive and negative fluxes. Furthermore, subalpine cold content was consistently lower in magnitude, meaning it took less energy input to drive cold content to zero and relative fluctuations were larger. Therefore, shallower snowpacks with reduced cold content, like those in the subalpine, are more susceptible to relatively 5 rapid changes in internal energy from surface energy fluxes.

### 5.3 Other controls on seasonal snowmelt timing and rate

Previous research has suggested uncertainty in the degree to which cold content controls snowmelt timing at daily to seasonal time scales. In our research, we found no statistically significant relationship between peak cold content magnitude and seasonal snowmelt onset using data from both observations and simulations. Rather, the majority of the variance in 10 seasonal snowmelt onset was explained by the timing of annual peak cold content and total spring precipitation. Later peak cold content generally occurred due to cold spring storms depositing significant snowfall. If such events were then followed by continued snowfall, then snowmelt timing was delayed. Meanwhile, seasonal snowmelt rate, or the ablation slope, was primarily controlled by peak SWE magnitude and timing, with greater, later peak SWE corresponding to more rapid snowmelt.

15 These results all suggest later seasonal snowmelt onset and faster snowmelt rates are primarily a function of persistent snowfall. While snowfall events can add significant cold content to the snowpack, they also change other fundamental properties that can delay snowmelt timing, such as increasing surface albedo (Clow et al., 2016) and adding dry pore space that must be saturated (Seligman et al., 2014). Additionally, other research suggests seasonal snowmelt onset is also affected by air temperature (Kapnick and Hall, 2012) and snow surface impurities (Painter et al., 2010; Skiles et al., 2012). Given the 20 importance of seasonal snowmelt timing to water resources management and various hydrologic processes, future work should focus on disentangling the effect of various physical processes on snowmelt rate and timing across snow-dominated regions globally, leveraging both field observations and snow model simulations.

### 5.4 Cold content development processes in other seasonal snow classes and climates

Despite the research presented here, there are still unanswered questions regarding cold content development as well as its 25 effect on snowmelt rate and timing. Firstly, we have only presented results from two sites within a single snow-dominated research catchment. Seasonal snow cover in the western United States spans a large elevational gradient and includes both maritime (e.g., the Cascades and Sierra Nevada) and continental (e.g., the Rocky Mountains) snowpack regimes (Serreze et al., 1999; Sturm et al., 1995). Therefore, an avenue for further research is to examine differences in cold content development across seasonally snow covered areas, with a particular focus on disentangling the effects of precipitation and 30 air temperature during snowfall at sites with different snowpack characteristics. For example, snowpacks in California's Sierra Nevada are typically deep, but air temperatures are generally near freezing even during winter storm events. Given the cold content of precipitation is a linear function of air temperature and precipitation depth (Eq. 2), a given unit of snowfall in



the Sierra Nevada should correspond to a lower cold content value than that same unit in the colder Rocky Mountains. Therefore, the control that precipitation exerts on cold content development is likely different between the two locations.

Secondly, a large amount of recent literature has shown unequivocally that, due to climate warming, patterns of snow accumulation and melt are changing across the globe with resultant effects on myriad hydrologic processes (Barnhart et al., 2016; Berghuijs et al., 2014; Knowles et al., 2006; Mote et al., 2005; Musselman et al., 2017; Pederson et al., 2011; Stewart, 2009). It is uncertain what role, if any, cold content plays in the climate-driven changes on snow processes. In our investigations we found pit-observed SWE was a strong predictor of cold content (alpine $r^2 = 0.84$; subalpine $r^2 = 0.50$), with subalpine cold content lower per unit SWE due to warmer depth-weighted snowpack temperatures. Both sites also exhibited a significant positive linear relationship between the cumulative mean of air temperature and snowpack temperature. Therefore, a unit of SWE in a warmer location or climate should correspond to reduced cold content due to increased snowpack temperature. Our work showed that decreased cold content magnitudes corresponded to faster snowmelt rates and earlier snowmelt timing at time scales less than 1 month. Therefore, reductions in snowpack cold content due to climate warming have implications for meltwater timing and availability, which could impact water resources management.

## 6 Conclusions

We have presented a long-term analysis of snowpack cold content using data from a long-term snow pit record and 23 y of physics-based snow model simulations at an alpine and subalpine site within the Niwot Ridge LTER. The research questions were designed to fill important missing gaps in the snow hydrology literature, namely the meteorological and energy balance processes behind cold content development and how cold content controls snowmelt rate and timing. To improve on the temporal resolution of the snow pit record, we ran the physics-based SNOWPACK model with a quality controlled, serially complete hourly dataset from WY1991–WY2013, a period covering a wide range of snowpack conditions.

Observations and simulations showed new snowfall was the primary pathway for cold content development, being responsible for 84.4% and 73.0% of modeled cold content gains in the alpine and subalpine, respectively. Snowfall days with cold content gains outnumbered dry days by a 4.2:1 ratio in the alpine and 2.6:1 in the subalpine. A negative energy balance—averaging $> -6.0$ W m$^{-2}$ in the alpine and subalpine—was responsible for the remainder of cold content gains, primarily due to the cooling effect of sublimation and net longwave emissions. At subdaily time scales, dry-period cold content increases occurred preferentially at night at both sites.

Seasonal snowmelt timing was not significantly correlated with peak cold content magnitude, but rather the timing of peak cold content and total spring precipitation controlled snowmelt onset. Later peak cold content and increased spring precipitation delayed snowmelt in both the alpine and subalpine, explaining 84.7% and 61.4% of the variance in peak SWE timing. Cold content magnitude did affect sub-seasonal snowmelt in that non-zero initial cold content values corresponded to delayed snowmelt timing and slower snowmelt rates. At daily time scales, the majority of melt events and the fastest melt rates occurred only when $CC_{6AM} = 0.0$ MJ m$^{-2}$. Any existing energy deficit at 6AM damped daily snowmelt rates.





The Niwot Ridge LTER provided the ideal study location for the research presented in this paper. The site's unique long-term snow pit and hourly meteorological records facilitated a high level of analysis into snowpack processes using both observations and physics-based snow model simulations. Lacking either data source would have limited the scope of this paper and added further uncertainty. Therefore, we hope this work underlines the utility of long-term *in situ* snowpack and

meteorological measurements as they allow for in-depth analyses on the observations themselves and also enable model validation on multiple snowpack properties (e.g., mass, depth-weighted temperature, and cold content), which improves the quality of simulated output.

**Data availability**

The quality controlled, infilled meteorological dataset presented in this work will be posted on the Niwot Ridge LTER

website (http://niwot.colorado.edu/index.php/data/). Please use this paper as the data citation and contact KSJ with questions (Keith.Jennings@colorado.edu). Snow pit (http://niwot.colorado.edu/index.php/data/data/snow-cover-profile-data-for-niwot-ridge-and-green-lakes-valley-1993-ongoi) and precipitation data (http://niwot.colorado.edu/index.php/data/data/precipitation-data-for-c1-chart-recorder-1952-ongoing and http://niwot.colorado.edu/index.php/data/data/precipitation-data-for-saddle-chart-recorder-1981-ongoing) can also be

accessed through the Niwot Ridge LTER. Niwot SNOTEL data can be found at http://www.wcc.nrcs.usda.gov/nwcc/site?sitenum=663. Niwot Ridge AmeriFlux data were provided by PI Peter Blanken and site manager Sean Burns and can be accessed at http://urquell.colorado.edu/data_ameriflux/.

**Appendix A**

**A.1 Meteorological data quality control and infilling**

The quality control routine for all observation types except precipitation followed the three-step procedure outlined in Meek and Hatfield (1994) where observations were flagged for removal if: 1) they fell outside of a prescribed minimum-maximum range for that day of year; 2) their hourly rate of change exceeded a given threshold; 3) the same value was recorded in four consecutive time steps, indicating a stuck sensor. A full description of the protocol for each variable falls outside the scope of this paper, but can be viewed in Meek and Hatfield (1994). The only changes made to their schema were applied to better

represent climate processes on Niwot Ridge, particularly the high variability in hourly air temperature and wind speed common at dry, high-elevation, mountainous, continental locations. These modifications allowed more valid observations to pass the quality control checks than the original Meek and Hatfield (1994) protocol.

Following the quality control procedure, missing observations were imputed using a hierarchical routine based on the work of Liston and Elder (2006), Kittel (2010), and Henn et al. (2012), where gaps of 72 h and shorter were infilled

using temporal techniques and longer gaps were infilled using a multi-station regression. Data gaps of 1 h were filled using a



linear interpolation between the observations directly preceding and following the missing value. Gaps between 2 h and 24 h were filled using an average of the value recorded 24 h prior and 24 h after the missing observation. Gaps between 25 h and 72 h were filled using a forecasted and back-casted autoregressive integrated moving average (ARIMA) model with imputed values linearly weighted by their temporal distance from the beginning/end of gap. Data gaps longer than 72 h, plus shorter

gaps that could not be filled using the temporal protocol due to missing data, were infilled with a one- or two-station regression. If both remaining stations were reporting valid observations, then the two-station regression was used. Otherwise, the one-station regression was employed. Regression equations were generated for each variable per month and 3 h time block where a day is divided into eight 3 h periods (e.g., 00:00–03:00, 03:00–06:00, etc.). Although such an approach neglects the spatial variability inherent to meteorologic processes in complex terrain, the values generated by the regressions

reproduce changes in conditions due to frontal passages and storm events. For periods when no stations were reporting, data were infilled using the mean value for the given station, variable, month, and 3 h time block.

       Quality controlled, gap-filled relative humidity, air temperature, and incoming solar radiation measurements were used to generate two estimates of incoming longwave radiation at an hourly time step. The equations presented in Angström (1915) and Dilley and O'Brien (1998) were used to estimate clear sky atmospheric emissivity based on vapor pressure,

which was calculated from relative humidity. Flerchinger et al. (2009) noted these two methods performed best at the subalpine site on Niwot Ridge relative to observations from the co-located AmeriFlux tower. Emissivity was then corrected for estimated cloud cover based on the ratio of observed solar radiation to maximum clear sky solar radiation using the approach of Crawford and Duchon (1999). Finally, incoming longwave radiation was calculated using the Stefan-Boltzmann equation:

$$LW \downarrow = \epsilon \sigma T_a^4 \qquad\qquad\qquad (A1)$$

where $LW \downarrow$ is incoming longwave radiation (W m$^{-2}$), $\epsilon$ is the estimated atmospheric emissivity (dimensionless, 0 to 1), $\sigma$ is the Stefan-Boltzmann constant ($5.67 \times 10^{-8}$ W m$^{-2}$ K$^{-4}$), and $T_a$ is air temperature (K).

       Measuring solid precipitation is inherently difficult, particularly at higher wind speeds (Rasmussen et al., 2012; Yang et al., 1999) and snowpack simulations are reliant on accurate precipitation input to produce reliable output (Raleigh et al., 2015; Schmucki et al., 2014). Thus, any snow modeling project has the compounded problem of requiring accurate

precipitation forcings and sensitivity to said forcings. For this study, two primary precipitation data sources were utilized along with site-specific gage corrections as described below.

       Alpine precipitation data came from the quality controlled LTER dataset (http://niwot.colorado.edu/index.php/data/data/precipitation-data-for-saddle-chart-recorder-1981-ongoing). While snowfall undercatch is commonly documented in the literature, Williams et al. (1998) showed blowing snow events lead to significant

overcatch at the LTER alpine precipitation gage from October through May. To correct the overcatch we created monthly precipitation reduction factors by comparing cumulative precipitation from the date of each snow pit observation to the following snow pit observation to the change in SWE between those observation dates when the change in pit SWE was positive. We found overcatch was greatest in months where Berg (1986) reported the highest frequency of blowing snow





events (January, March —average reduction = 0.59) and lowest in months with fewer blowing snow events (December, February, April—average reduction = 0.86).

Subalpine precipitation data came from the quality controlled, gap-filled Kittel et al. (2015) dataset with further corrections applied for snow undercatch relative to the Niwot SNOTEL snow pillow during snowfall events, which averaged
2.1 mm per snowfall day. Air temperature during precipitation events showed the strongest control on undercatch with decreasing air temperature corresponding to increased negative precipitation biases. Notably, wind speed was not correlated with undercatch at the subalpine gage, likely due to the siting of the anemometer. This instrument is located 5 m above ground level in a roadside clearing and is generally unrepresentative of the wind speed magnitude in the dense subalpine forest where the snow pit, LTER precipitation gage, and Niwot SNOTEL station are located. Compared to the subalpine
snow pit, accumulated precipitation in the gage was on average 88.3 mm or 32.3% lower than observed maximum SWE.

Daily precipitation observations from both datasets were temporally disaggregated to the hourly time step of SNOWPACK by dividing the daily total by 24 and equally distributing the values to each hour of the day. Hourly precipitation observations were not available, and therefore a more advanced disaggregation method was not pursued.

### A.2 Meteorological data infilling validation

Missing observations and measurements failing the quality control checks were more common in the alpine than subalpine (Table A1). The variable with the greatest number of missing values was solar radiation in the alpine due to a long instrument outage period in the 2000s. The multi-station regression was the most utilized infilling technique (temporal infilling accounted for, at most, 3.0% of the missing data) and cross-validation statistics are presented in Table A1. Generally, infilling performance was greater in the alpine due to the close proximity of the high alpine meteorological
station. Of the forcing variables, air temperature exhibited the highest infilling performance and wind speed the lowest.

Estimates of incoming longwave radiation exhibited low biases relative to shorter-term observations taken near the alpine and subalpine meteorological stations. In the alpine, measurements of incoming longwave radiation were taken at the Subnivean Laboratory from 1996 through 2008 and intermittently in more recent years. Here, the Dilley and O'Brien (1998) equation produced the best results relative to the observed data with a mean bias of 4.9 W m$^{-2}$. In the subalpine, the mean
bias relative to Ameriflux observations (1999-07-12 through 2013-12-31) was 10.4 W m$^{-2}$ with the Angström (1915) estimate providing the best match. The positive biases in the alpine and subalpine represented 2.0% and 4.1%, respectively, of the average hourly observed incoming longwave radiation, values which were within the manufacturer-reported precision range of ±10% for the Kipp and Zonen CG2 net pyrgeometer at the Subnivean Laboratory and the CNR1 net radiometer at the AmeriFlux tower. The coefficient of determination for hourly and daily incoming longwave values were 0.51 and 0.72,
respectively, in the alpine and 0.44 and 0.60 in the subalpine.





**Author contributions**

KSJ, TGK, and NPM designed the study. KSJ performed the analyses and wrote the manuscript. TGK and NPM provided feedback and edited the manuscript.

**Acknowledgments**

KSJ was supported by a NASA Earth and Space Science Fellowship (16-EARTH16F-378). The Niwot Ridge LTER is funded by the United States National Science Foundation (NSF DEB #1637686). We are grateful to Mark Williams for creating and sustaining the Niwot Ridge LTER Snow Internship Program, from which the snow pit data were derived. We also offer our thanks to Ben Livneh and Andrew Badger who provided feedback on the infilling procedure, as well as to Hope Humphries and Jennifer Morse who helped track down missing meteorological data.

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

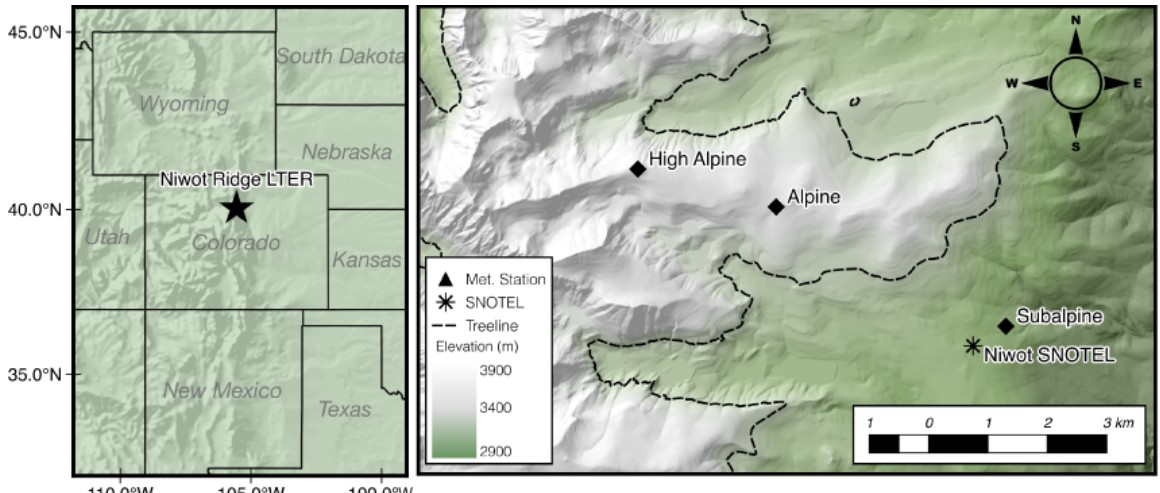

10   **Figure 1: The Niwot Ridge LTER and surrounding environment. The dashed line in the LTER inset represents approximate treeline (3400 m). The snow study focused on the alpine and subalpine sites, the two locations which have co-located snow pit observations and meteorological stations. The high alpine site was used as an additional station in the meteorological data infilling protocol and the Niwot SNOTEL was used for model validation.**





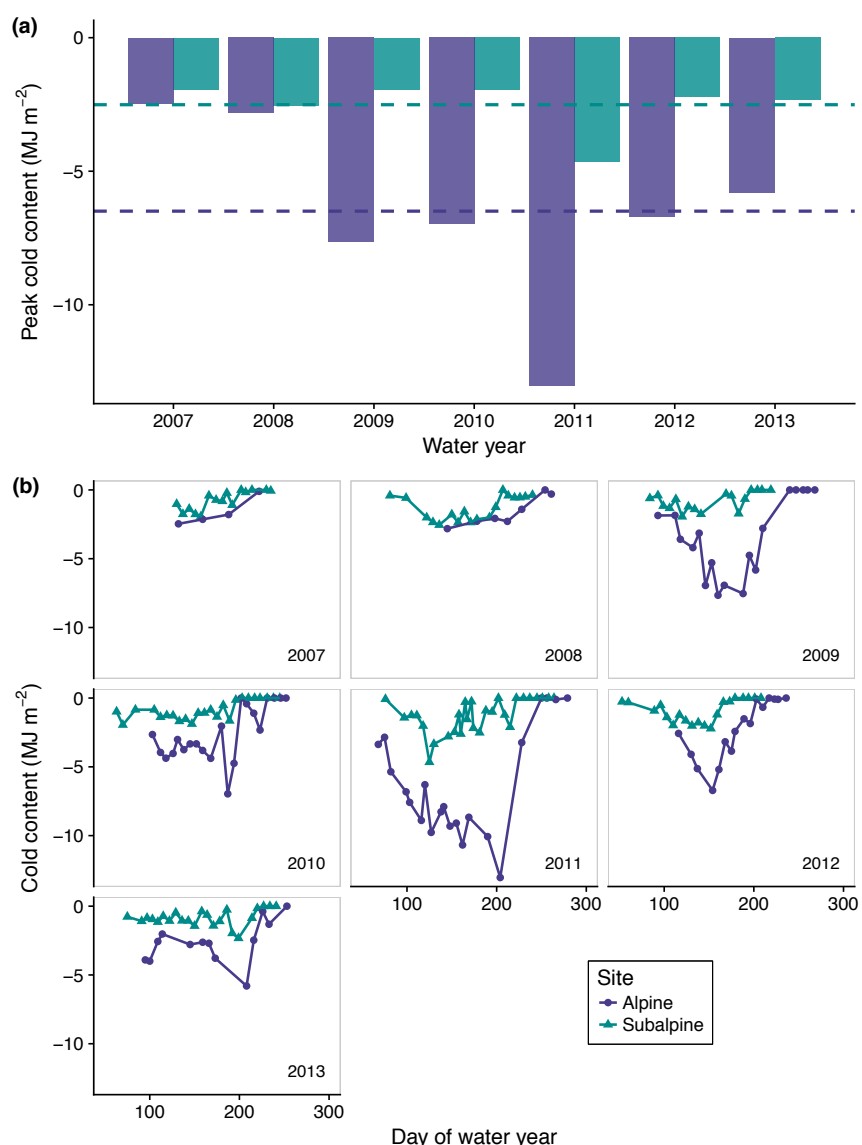

**Figure 2. Peak annual cold content (a) and individual snow pit observations of cold content (b) for the alpine and subalpine from WY2007–WY2013. The dashed horizontal lines in (a) represent the mean peak annual cold content values for the two sites.**



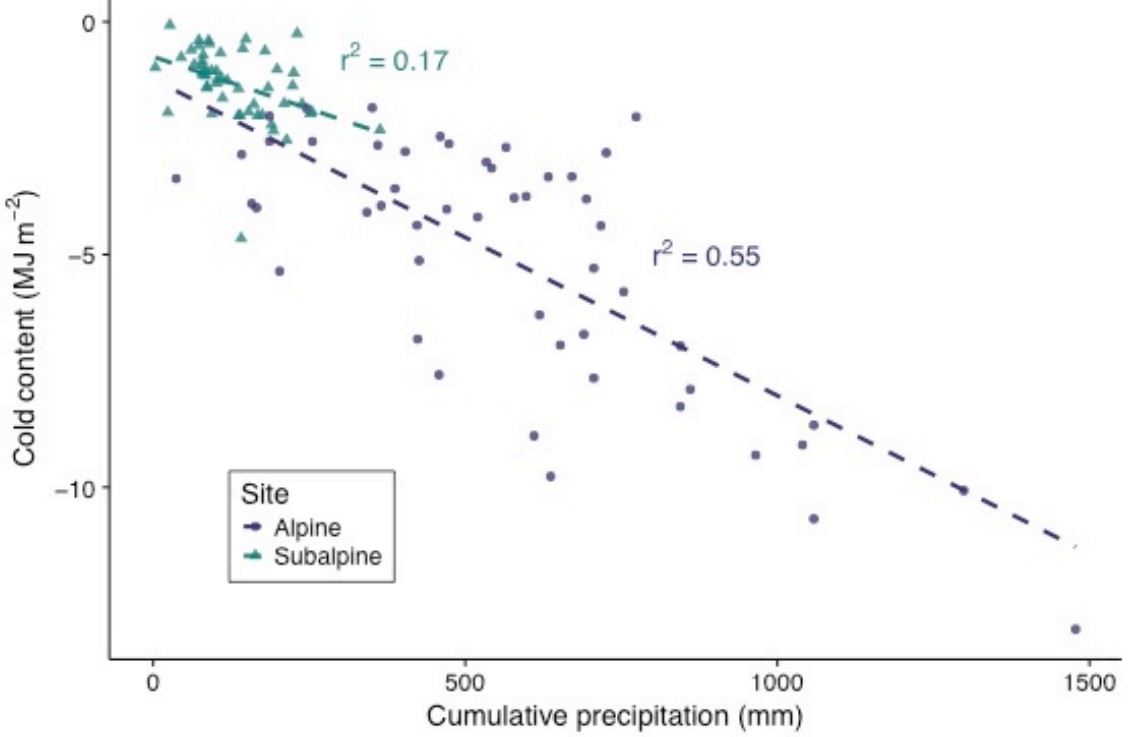

**Figure 3. Cold content plotted against cumulative precipitation from 1 December to the date of snow pit observation for the alpine and subalpine for the snow season up to including the date of peak cold content from WY2007–WY2013. The dashed lines of best fit were calculated using ordinary least squares linear regression.**





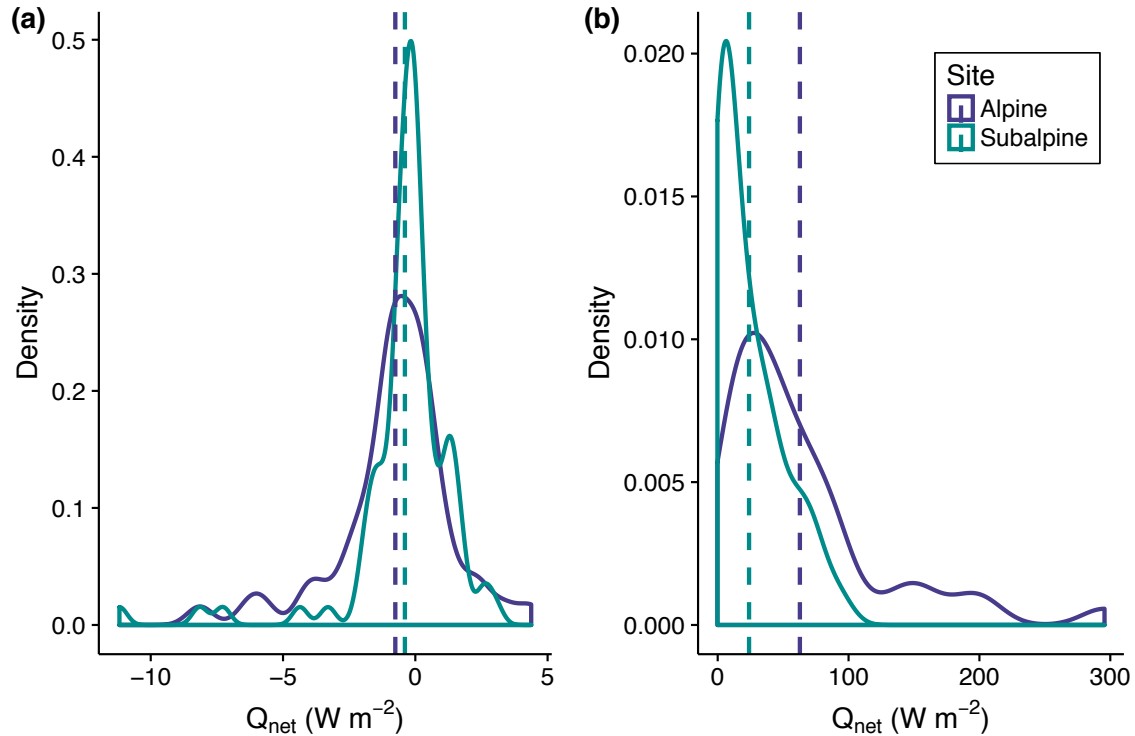

**Figure 4. Kernel density estimates of $Q_{net}$ distributions as calculated from snow pit observations for periods with SWE gain (a) and loss (b) in the alpine and subalpine for WY2007–WY2013. The dashed vertical lines represent the mean $Q_{net}$ for the alpine (a = -0.8 W m$^{-2}$; b = 62.8 W m$^{-2}$) and subalpine (a = -0.4 W m$^{-2}$; b = 23.9 W m$^{-2}$).**





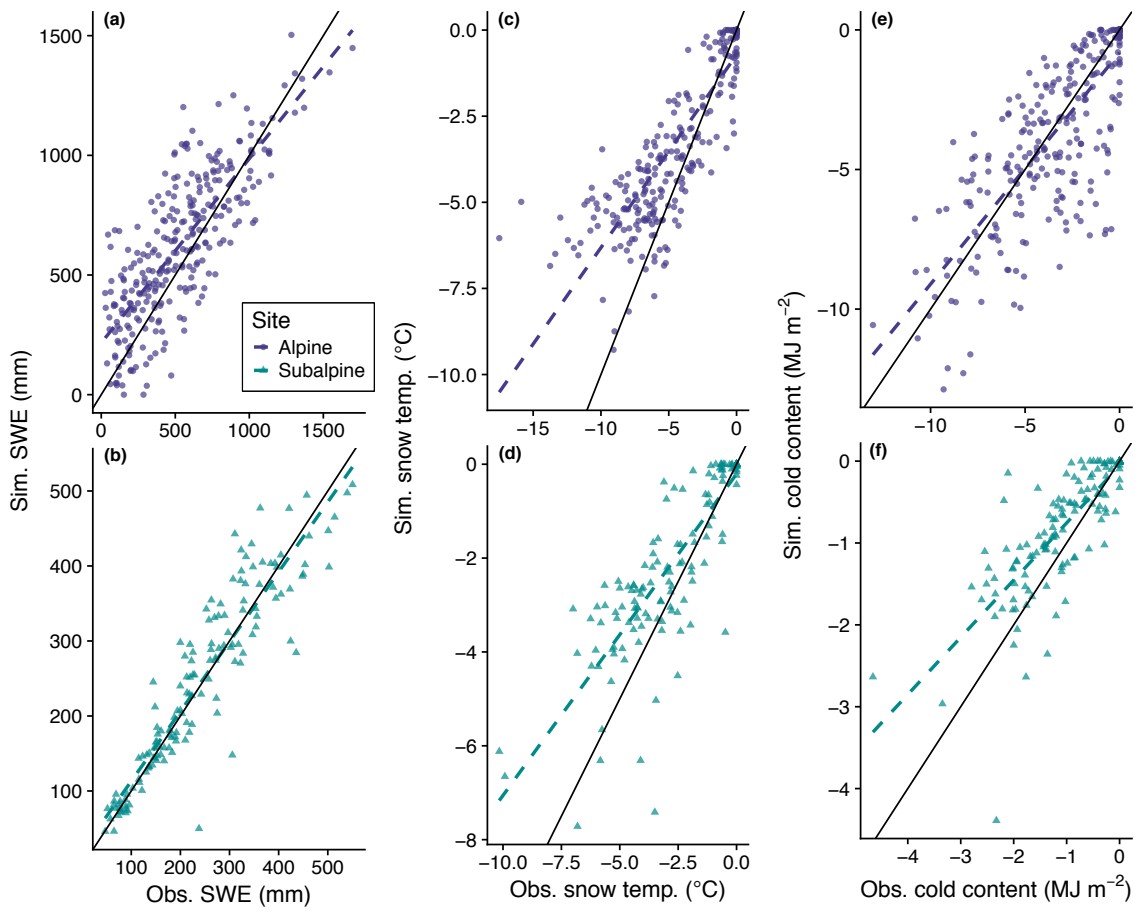

**Figure 5.** Plots of simulated versus snow-pit observed SWE (a,b), snowpack temperature (c,d), and cold content (e,f) in the alpine (top, WY1995–WY2013) and subalpine (bottom, WY2007–WY2013). The solid black line is the 1:1 line and the dashed lines are the lines of best fit as determined by ordinary least squares linear regression. Simulation error metrics are presented in Table 1.



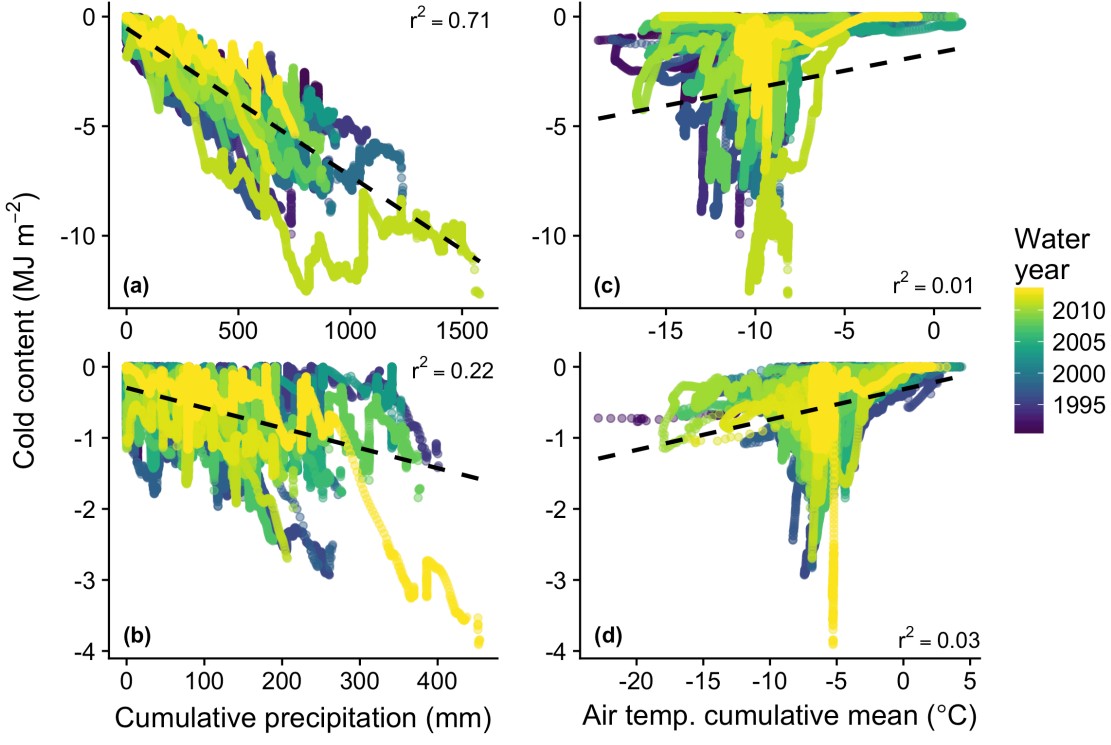

Figure 6. Simulated cold content plotted against cumulative precipitation in the alpine (a) and subalpine (b), and the cumulative mean of air temperature in the alpine (c) and subalpine (d). Shading denotes the corresponding water year.





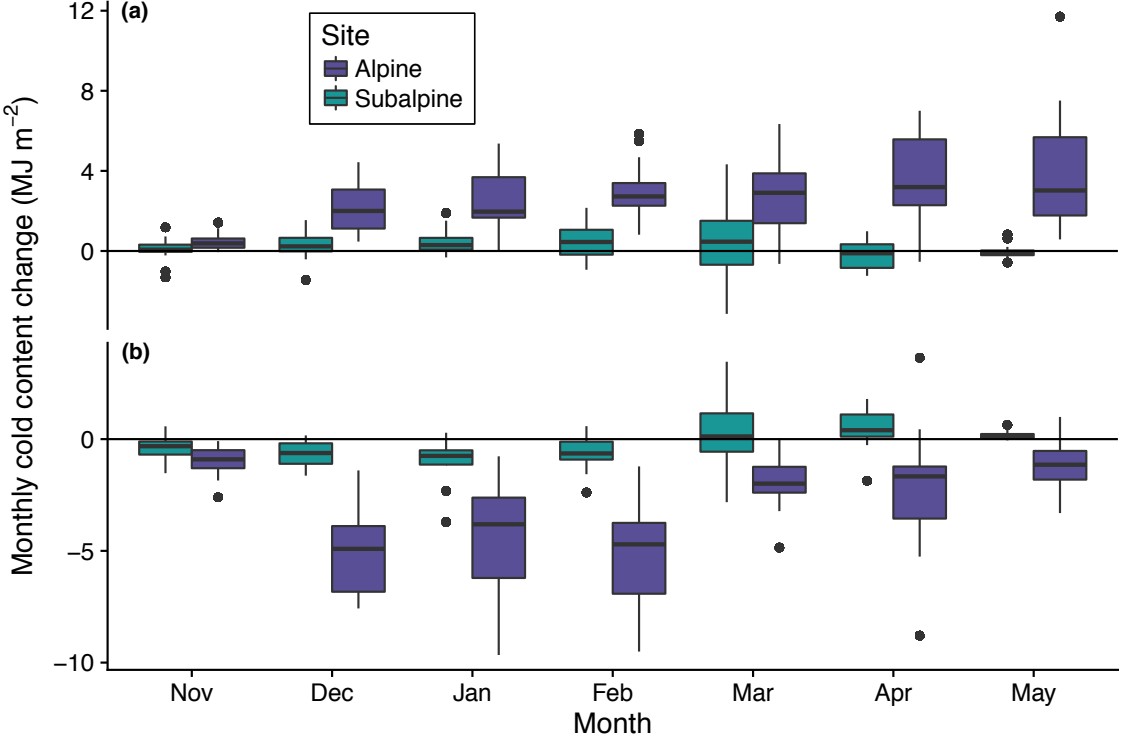

**Figure 7. Simulated cold content gain and loss per month in the alpine and subalpine for dry days (a) and wet days (b). Values above the zero line correspond to a loss of cold content (i.e., cold content approaches zero), while values below correspond to a gain of cold content.**





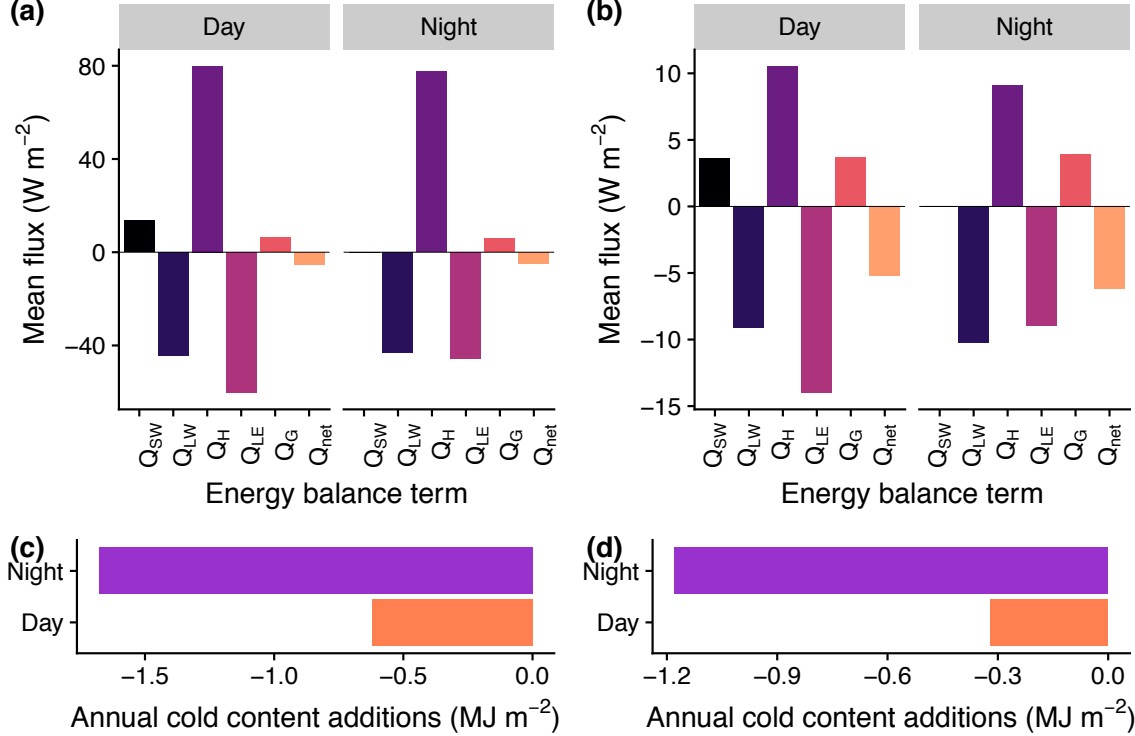

**Figure 8.** Simulated snowpack energy balance during the day (0600 h to 1800 h) and night (1800 h to 0600 h) in the alpine (a) and subalpine (b) for periods of cold content gain without precipitation, plus total cold content contributions during day and night periods in the alpine (c) and subalpine (d). Note: In (a,b) $Q_R$ is not shown because rain-on-snow events are rare at both sites and they also do not contribute to cold content gains (i.e., rain advects energy to the snowpack).





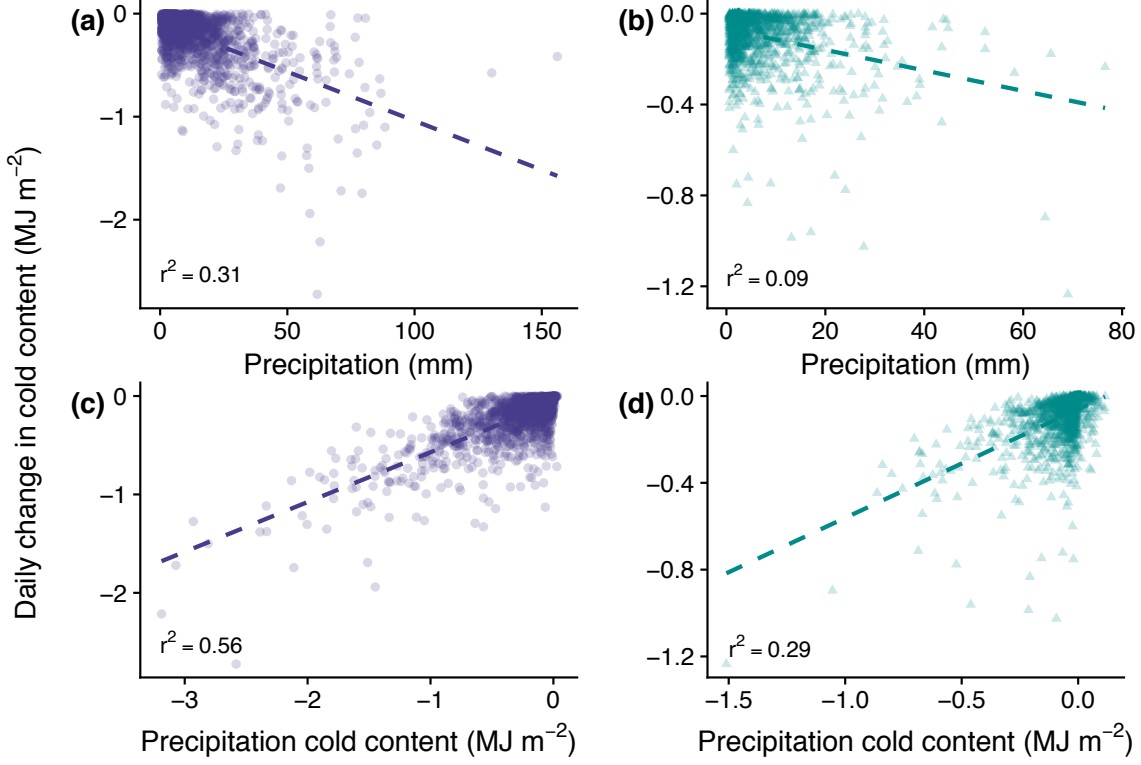

**Figure 9.** Simulated daily change in cold content plotted against daily precipitation in the alpine (a) and subalpine (b), and cold content from precipitation in the alpine (c) and subalpine (d).



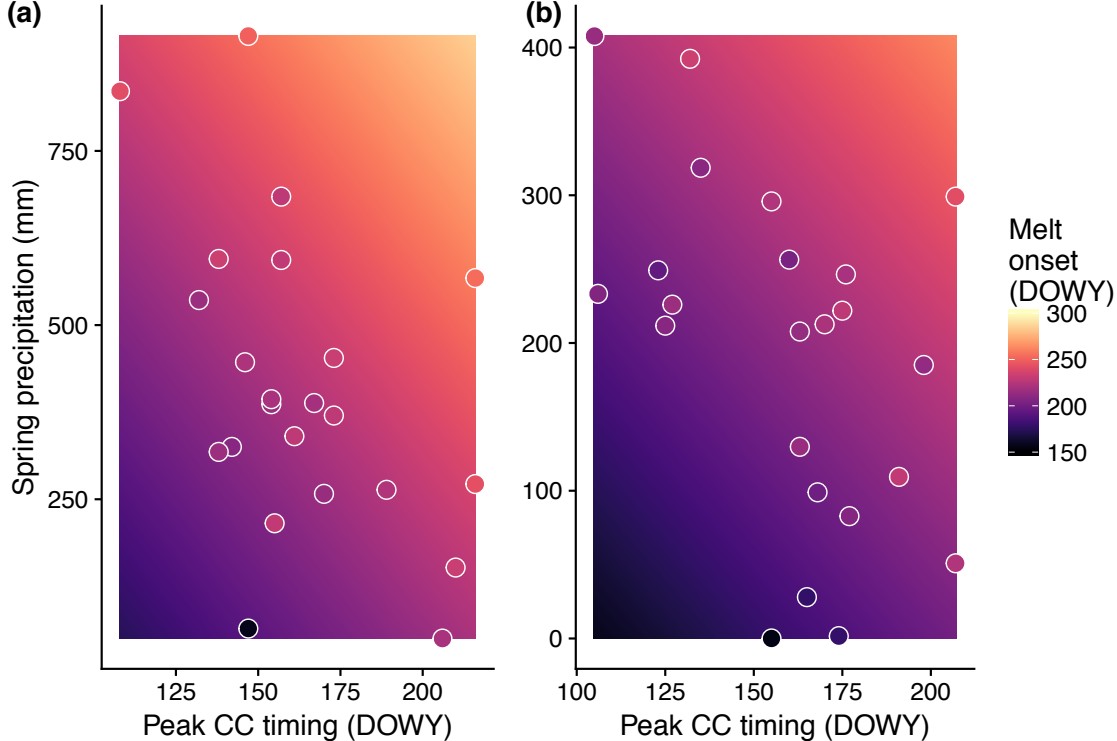

**Figure 10. Annual melt onset as predicted by peak cold content timing and spring precipitation in the alpine (a) and subalpine (b). The background gradient in each plot displays the predicted melt onset DOWY as calculated by a multiple linear regression, while the shading within each point represents the actual melt onset simulated in a given water year at its peak cold content timing DOWY and spring precipitation value. At both sites melt onset is delayed by later peak cold content and increased spring precipitation.**



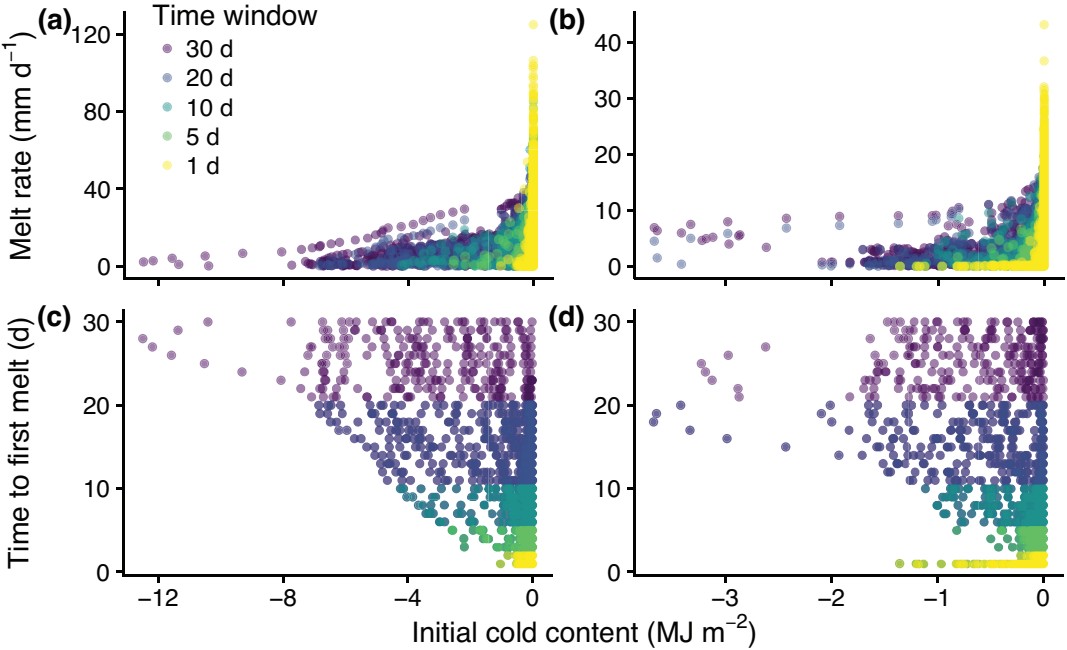

**Figure 11.** Simulated sub-seasonal snowmelt rate plotted against initial cold content in the alpine (a) and subalpine (b), and time to first melt plotted against initial cold content in the alpine (c) and subalpine (d) for time windows from 1 d to 30 d.





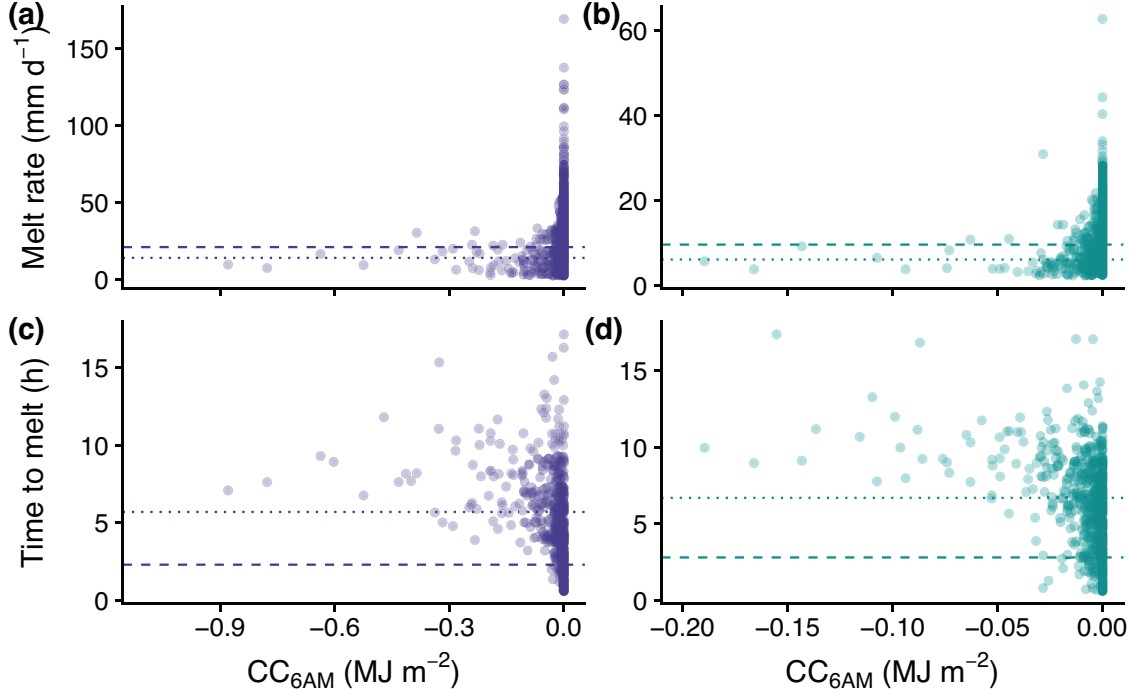

**Figure 12.** Simulated daily melt rates in the alpine (a) and subalpine (b) and time to snowmelt in the alpine (c) and subalpine (d) as a function of $CC_{6AM}$. The dashed line in each figure represents the mean melt rate (a,b) and time to melt (c,d) for days when $CC_{6AM} = 0$ MJ m$^{-2}$ and the dotted line represents those quantities for days when $CC_{6AM} < 0$ MJ m$^{-2}$.





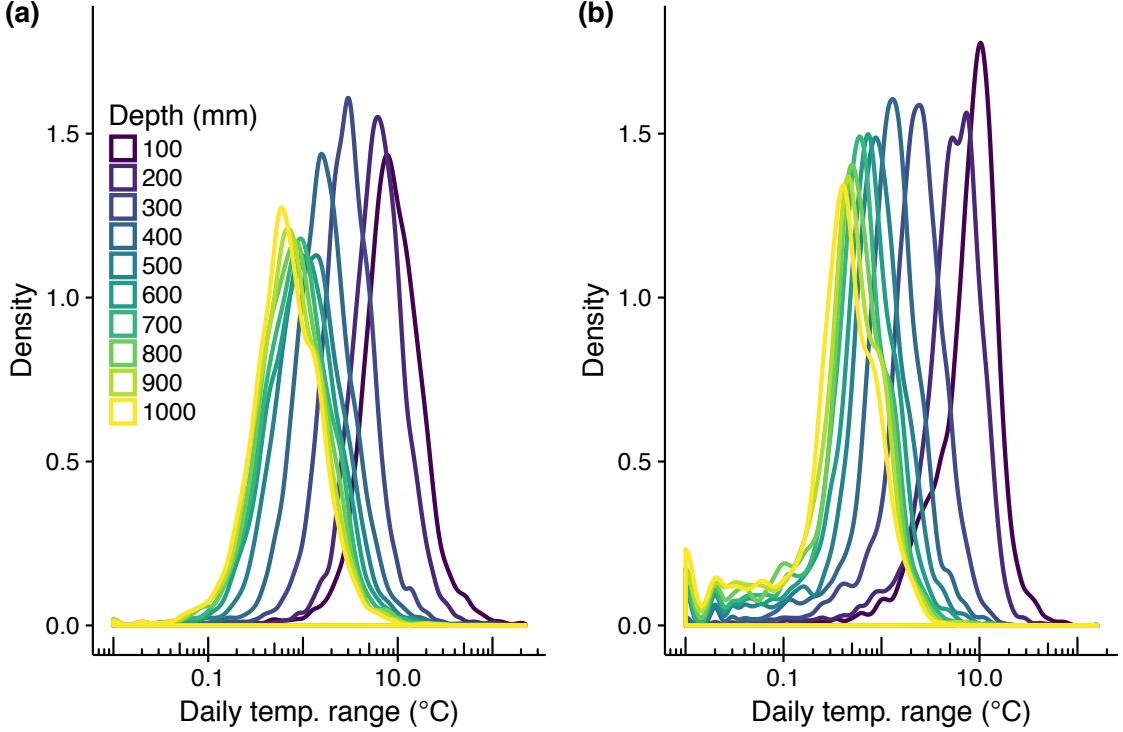

**Figure 13. Kernel density estimates of simulated daily snowpack temperature ranges in the alpine (a) and subalpine (b). Line shading represents the bottom depth of the layer with layers near the top of the snowpack in purple and blue and lower layers in green and yellow.**



**Table 1. Mean quantities for the alpine and subalpine snow pits from WY2007–WY2013**

| Site | Peak CC (MJ m$^{-2}$) | Peak SWE (mm) | Date of Peak CC | Date of Peak SWE |
|---|---|---|---|---|
| Alpine | -6.5 | 843 | 19-March | 6-May |
| Subalpine | -2.5 | 395 | 14-February | 26-April |

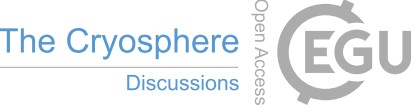



**Table 2. Statistics for SNOWPACK simulations relative to daily and annual observations from the snow pits in the alpine and subalpine, and Niwot SNOTEL in the subalpine. There is no SNOTEL station in the alpine and SNOTEL does not observe cold content and snowpack temperature. Comparisons are for the water years listed in the second column.**

| Site | WY Range | Daily | | | | | | Annual | |
|---|---|---|---|---|---|---|---|---|---|
| | | SWE $r^2$ | SWE Mean Bias (mm) | $T_s r^2$ | $T_s$ Mean Bias (°C) | CC $r^2$ | CC Mean Bias (MJ $m^{-2}$) | Max SWE Mean Bias (mm) | Max CC Mean Bias (MJ $m^{-2}$) |
| Alpine | 1996-2013 | 0.63 | 95.8 | 0.74 | 1.1 | 0.63 | -0.3 | 99 .0 | -0.7 |
| Subalpine (Snow Pit) | 2007-2013 | 0.85 | 3.4 | 0.72 | 0.6 | 0.63 | 0.2 | 15.0 | 0.6 |
| Subalpine (SNOTEL) | 1991-2013 | 0.89 | -5.4 | NA | NA | NA | NA | 44.1 | NA |





**Table A1.** Cross-validation statistics for the multi-station regression infilling procedure for air temperature ($T_a$, °C), total incoming solar radiation ($SW_{in}$, MJ m$^{-2}$), wind speed (VW, m s$^{-1}$), and dew point temperature ($T_d$, °C). Note: Relative humidity values were converted to $T_d$ for computing the multi-station regression.

| Site | Variable | Missing Obs. (%) | Mean Bias | RMSE | $r^2$ |
|---|---|---|---|---|---|
| Alpine | $T_a$ | 8.2 | $2.8 \times 10^{-3}$ | 1.6 | 0.97 |
| | $SW_{in}$ | 25.3 | $-4.4 \times 10^{-2}$ | 0.4 | 0.83 |
| | VW | 6.0 | -0.5 | 3.2 | 0.69 |
| | $T_d$ | 6.9 | -1.3 | 3.7 | 0.84 |
| Subalpine | $T_a$ | 3.8 | $-6.4 \times 10^{-2}$ | 3.5 | 0.86 |
| | $SW_{in}$ | 2.9 | $-4.8 \times 10^{-2}$ | 0.6 | 0.67 |
| | VW | 3.6 | -0.3 | 2.1 | 0.30 |
| | $T_d$ | 3.6 | -2.9 | 4.7 | 0.81 |