# Peer review of "Observations and simulations of the seasonal evolution of snowpack cold content and its relation to snowmelt and the snowpack energy budget"

_The Cryosphere, 2017_

## Referee Comment (RC1) · Anonymous Referee #1 · 22 Dec 2017

The authors present a study that uses a long term observational data set to validate simulated snowpack cold content. The authors attribute the largest increase in cold content to new precipitation mass. Validating a complex, multi-layer snowpack model that is frequently used in the literature is a substantial contribution, especially given the uniqueness of the long-term snow pit data. However, as currently presented, this manuscript needs substantial revision and polish. Below I explain my reasoning for this, and I hope the authors can use it to improve this manuscript into the contribution that is hiding under the surface. As is, I recommend accept pending major revisions.

My first issue is that these conclusions are specific to a deep snowpacks in a warmer climate. Thin, shallow snowcovers have a long record in the literature as being difficult to simulate due to the substantial radiative cooling of the snowpack resulting in sharp gradients and maximum cold content being exceeded. It is important that all these results are very clearly stated to apply to the deep snowpacks herein.

Second, is that I'm not entirely convinced by the results. As I understand it, the authors assert via Figure 3 that cold content of the snow pack is explained by cumulative precipitation. A statistically significant trend line is show for the subalpine site; however, it has an r^2 of 0.17. Cold content is effectively an instantaneous, integrated snowpack temperature expressed as energy required to bring it to zero-degree isothermal. Cold content will, by definition, become greater (more negative) as below zero-degree mass is added to the snowpack. An r^2 of 0.17 is a poor correlation and does not, at least to me, act as strong evidence for the authors conclusion. Perhaps the r^2 for the alpine site is acceptable, however given cold content will by definition increase as cold mass is added, it seems to be a circular result that does not add any new knowledge nor should be unexpected. With these results, the authors then proceed to the model step, effectively trying to duplicate the observed results. Stepping back, the message I feel like the authors are trying to present are: "there is no substantial radiative cooling of the snowpack, thus the precipitation temperature (and associated cold content) is the principal control on the total snowpack temperature, and therefore cold content." I suspect this is where Figure 8 becomes important, showing a small, negative total Qnet. However, something feels off about these results. In Figure 8a, the only real difference between day and night is the shortwave radiation and a slightly dampened latent heat flux. It seems odd to me that the mean response is identical, especially for the sensible heat flux. I'm just highly skeptical of an almost entirely similar surface energy balance between night and day. I would like the authors, upon confirming these results are correct as presented, to describe in more detail what is going on here, and if this is a site-specific effect or not, as my impression is it may be. Stepping back to Figure 6, I feel like this further highlights my issue with this conclusion. Full energy balance

models use the balance of the energetics to simulate internal layer temperatures and energetics. Using cumulative mean air temperature feels very temperature-indexy and not really appropriate in this context – it supposes that the entirely of the snowpack energetics could potentially be explained by a mean air temperature, when in reality it's really the associated processes that would impact it.

Third, precipitation temperature and phase is unaddressed and is a critical component of this work. The simulations shown in Figure 9 c and d suppose the precipitation temperature and phase are correct. I'm assuming you used the default temperature-threshold in Snowpack for phase? These results could be quite different if phase was wrong (i.e., rain instead of warm snow) or precipitation temperature was biased. There is substantial uncertainty associated with phase partitioning methods and snowfall temperature (e.g., Harder, et al. 2014), and these have significant implications for this work. How sensitive are these results to various phase and falling hydrometeor temperatures?

Fourth, despite reading through this a few times looking for it, it is unclear to me what kind of clearing this sub-alpine site is in. The site is specifically stated as a clearing, but the Snowpack canopy routine is enabled. This will significantly change the surface fluxes as well as precipitation at the snow surface; e.g., canopy interception. In my mind, this undermines the results presented herein – maybe it explains the poor result in Figure 3? – and needs to be detailed and the effects and impacts explained. Site photos would go a long way towards helping orient the reader. However as is, this is a major detail that is omitted.

Fifth, A discussion on the role of $Qg$ on cold content is needed and the assumptions behind your $Qg$ simulation flux. These results show a treatment of the surface fluxes on cold content, but neglect discussion of soil-snowpack interactions, e.g., conditions that lead to frozen soil or refreezing of active layers.

Lastly, the authors assert that increased peak cold content and total spring precipitation

control snowmelt onset. But this seems by-definition – doesn't this imply more mass and refreshed albedos? Isn't this just what you'd expect with increased cold content being a function of snowpack mass?

In summary: As I understand the results presented, the story is that the authors found limited evidence for sustained energy loss from the snowpack and that the cold content of the snowpack was mostly a result of mass inputs. However, there are many confounding factors that make it difficult to accept this at face value. Given the circular reasoning in the results (more snow -> more cold content, but that is by definition), it is difficult for the reader to accept the results. That being said, validating the model against these observations is quite interesting and diagnosing snowpack energy loss during the winter is a useful contribution. However, I think the overall message needs to be refined to more clearly articulate the site-specific nature of this study, the uncertainties in key aspects of the analysis (e.g., precipitation, canopy), and the text improved for readability.

References Harder, P., and J. W. Pomeroy (2014), Hydrological model uncertainty due to precipitation-phase partitioning methods, Hydrol. Process., 28, 4311–4327, doi:10.1002/hyp.10214.

Specific points Throughout: The authors introduce (para 25) increase/decrease for cold content, but proceed to use gain/loss. I think it should be consistent throughout Figure is used in the text but Fig. when used in brackets. Ideally should be consistent. Units should be separated with a cdot instead of spaces, e.g., WâŃĚmˆ(-2) Unclear what wet and dry days mean. Wet implies rain to me, but I suspect that's not what you mean. I would reword, or at least clearly define.

P1, Para 20: "cold content ... associated with reduced snowmelt" this needs to be reworded as snowmelt should be happening when CC = 0. Which melt rate is being considered? P2, Para 20: "the authors" which authors? P2, Para 25: "Furthermore. . ." I'm not sure I agree with this statement. CC needs to be = 0 for melt to occur, so isn't

this known? Do you have a citation? P2, Para 30: "saturate", word choice P2, Para 30: "However...", I'm unclear what you're trying to say, please clarify. P3, Para 10, 15, 30 Need to be indented. P3, Para 20, use "10 m/s to 13 m/s" instead of how it is written. P4, Para 1, "Snow" incorrect capitalized P4, Para 14, "downwelling longwave" I would put a quick note as to what method you used. P5, Para 20, remove "proposed in Sect. 1" P5, Para 20 "We then quantified" I found this section unclear P5, Eqn 3 Consider writing 86,400 as a variable and showing in the text the units. Either way, you need units. P5, Para 15 "in order to improve" Using a model doesn't improve obs, it just compliments them. I think you should reword to make this distinction. P5, Para 20 "number of finite elements" change to layers P5, Para 25 remove "the numerical model in" P5, Para 5, the canopy module stuff comes out of nowhere, especially given you say the site is in a clearing. This needs to be much clearer. P5, Para 20 "Output from snow model simulations" I don't follow. Do you mean the comparison is more robust w/multiple outputs to validate? P6, Para 20 Any EC observations considered? P7, Eqn 4 The form for the energy balance equation given in Equation 4 is not a standard form. Generally, the change in internal energetics are given as a dU/dt and Qm is on the LHS. Qnet and Qm together are redundant in the energy balance as the energy available for melt is the net energy. P7, Para 1, "time scales" -> temporal scales P7, Para 25, as I said above, I don't buy that an r^2=0.17 demonstrates a primary control P8, Para 5, Probably should note these are depth averaged P8, Para 10, -2.2 should have units after it P8, Para 15, How is this working with the canopy module? Intercepted snow has massive sublimation losses, but that doesn't seem to be reflected here. P8, Para 20, Monotonically is either monotonic or not. There is no in-between. Reword P8, Para 25, "simulations confirm" change to "support" or similar P9, Para 10, So how are you calculating Qg? Maybe I missed it? I think you need a reasonable treatment on the assumptions behind however you do this. Did you couple snowpack with the soil? Constant flux? Constant ground temp? Qg is important for a conduction heat flux into the snow pack, and needs to be addressed if you go after cold content. Often Qg is taken to be 0-4W/m^2, but this flux can be important for stopping a numerical
model from simulating absurd cold contents. P12, Para 10 "continued snowfall" But this is just more mass, so you'd expect snowmelt timing to be delayed P12, Para 20, "future work. . ." Lots of work on this already. . .. P16, Para 30 Given Snowpack is forced hourly, this longwave estimate seems like a massive source of uncertainty, especially within the context of an energy balance model. There are many incoming longwave formulations that take into account various proxies for non-clear sky. You seem to do this for your emissivity, but it's not clear how that exactly works. With such low r^2 this needs to be detailed and expanded upon. The large error in a critically important mid-winter energy flux may have substantial implications for this work.

Figures

All figures – It would certainly aid readability to have them labeled as alpine/sub alpine without having to constantly refer to the caption.

Figure 1, difficult to determine differences at high elevation. Figure 2, can you change the DOY to dates for easier parsing? Figure 5a,b Should have same axis extents Figure 8abcd would benefit from having the same y- (ab) and x- (cd) axes to aid in comparison. Also, please expand the y-axes of (ab) so-as to understand what the limits are. Figure 9, needs legend

---

## Referee Comment (RC2) · Anonymous Referee #2 · 1 Feb 2018

**Review of :**

**Observations and simulations of the seasonal evolution of snowpack cold content and its relation to snowmelt and the snowpack energy budget,**

**by Jennings et al.**

The authors address the issue of the drivers of cold content evolution based on the exemple of two seasonal snowpacks from a single observation catchment in the Western US. They also assess, in a much shorter part, the effect of cold-content on snow-melt timing and rates.

The paper is well written and well illustrated. The take-home message is clear and the objectives assessed in Introduction are achieved with scientific quality. I found the paper both an appreciable synthesis of existing litterature on the topic, and enlightning regarding the conclusions achieved.

In addition to the few suggestions made below, there is in y opinion one minor contradiction in the Result section (point 9 below), that I would recommand the authors to adress with priority as it affects the consistency of the paper. Note that this apparent contradiction may come from misunderstanding from my side, or an edit mistake from the side of the authors.

1. Introduction – p2 L1-6 : not the snowmelt itself is critical for the cited applications, but the timing of surface/subsurface runoff from Snow melt, which may not be the same when surface melt occurs and refreezes (as just mentionned earlier in the manuscript). For consistence I suggest changing « snowmelt » into « runoff from snowmelt » in the current sentence.

2. Introduction – p2 L16-18 : Aren't the « dominant processes » well-resolved by Method-3 (residual of the energy balance) at places where energy-balance models like SNOWPACK are routinely validated against data regarding most components of the energy balance ? In that case, isn't it rather a lack of investigation into the process of cold content development, than a lack of validation data, that limits knowledge of the prevailing processes ?

   Furthermore, the dominant processes involved in cold content development likely depend on the climatology of the investigated sites. Event though it is well stated in the Discussion, specifying the perimeter of validity of your study should be done here already. Typically, I suggest transforming research question #1 (p3 L6) into «  What are the meteorological and energy balance controls on cold content development **at two alpine and sub-alpine sites from the Western US** ? »

3. Introduction – p2 L22-26 : conduction fluxes within the snowpack, and from snowpack to the ground, can mitigate the impact of the intense negative fluxes reported here. If a gradient around 100 W/m develops within the snowpack as a result of intense surface cooling, around 20 W/m2 propagates downwards (upon hypothesis of a 0.2 W/m/K conductivity for Snow), which should somewhat prevents the snowpack from locally reaching unreaslitic temperatures ( ?)

4. Introduction – p3 L1 : uncertainties -> unknowns (suggestion)

5. Methods – p5 L28 : « vapour diffusion » in SNOWPACK is actually only calculated to compute Snow grain/bounds growth rates. There is no mass redistribution between different snow layers as a result of vapour diffusion in current versions of SNOWPACK. I therefore suggest to suppress this item from the list of existing SNOWPACK routines, as it would be otherwise misleading.

6. Methods – p6 L7-10 : could you specify here or in appendix the result of your calibration procedure for the parameters leaf area index, vegetation height, direct canopy throughfall, and wind speed reduction ? Note that these parameters are usually estimated from field data, and that any observation-based estimate of them would help assess the soundness of the calibrated parameter or of the canopy model.

   Additionnal, the rough size of the clearing where sub-alpine snowpits were made, should be specified (p4 L1) to justify the use the canopy module of SNOWPACK, instead of an open-area version SNOWPACK with just wind attenuation.

7. Results – p5 L15-16 : « Peak cold content and peak SWE respectively occurred 33 d and 10 d later in the alpine than subalpine ». Add « on average » to this sentence and the next.

8. Results - p7 L25-27 : « This is likely due to the increased variability of winter precipitation, the coefficient of variation of which is 2.9 and 2.7 times greater than that of air temperature in the alpine and subalpine, respectively ». I assume that by « increased » you mean « higher » ? I would suggest that snow-atmosphere heat transfers occuring during cold air temperatures periods are less efficient in cooling the snowpack, than the direct addition of cold Snow from fresh snowfall.

9. Results - p7 L29-30 : « During periods of SWE accumulation, $Q_{net}$ was typically near 0 W m$^{-2}$ (Fig. 4a), indicating a large negative energy balance was not responsible for cold content development. » First, here, you infer $Q_{net}$ from the variation in CC between 2 snowpit dates, so where is the link to energy balance ? Second, based on Eq 3, $Q_{net} \sim 0$ W m$^{-2}$ indicates no cold-content increase, meaning there is no visible snowfall-driven cold-content increase in the snowpit data. In my mind this contradicts the other results of the study, e.g. Fig 3 and 7 – please justify, or explain me where I am wrong. May I suggest using different names for $Q_{net}$ in Eq. 3 and $Q_{net}$ in Eq. 4 ? Like $Q_{net-pit}$ and $Q_{net-EB}$ respectively.

10. Results - p8 L7 : could overestimated densities be the reason for cold-content overestimation at alpine location ? (as Snow temperature tend to be overestimated ?) Maybe a line on that could be added to the Result or Discussion section.

---

## Author Comment (AC1) · 7 Mar 2018

The authors present a study that uses a long term observational data set to validate simulated snowpack cold content. The authors attribute the largest increase in cold content to new precipitation mass. Validating a complex, multi-layer snowpack model that is frequently used in the literature is a substantial contribution, especially given the uniqueness of the long-term snow pit data. However, as currently presented, this manuscript needs substantial revision and polish. Below I explain my reasoning for this, and I hope the authors can use it to improve this manuscript into the contribution that is hiding under the surface. As is, I recommend accept pending major revisions.

Thank you for the detailed, critical review and the suggestion to publish pending the revisions. In the below response, our comments are in blue text.

My first issue is that these conclusions are specific to a deep snowpacks in a warmer climate. Thin, shallow snowcovers have a long record in the literature as being difficult to simulate due to the substantial radiative cooling of the snowpack resulting in sharp gradients and maximum cold content being exceeded. It is important that all these results are very clearly stated to apply to the deep snowpacks herein.

Given our geographic setting (Colorado Rocky Mountains), we framed this work in the context of western US snowpacks, which are essential to regional water resources. In this regard, the alpine and subalpine snowpacks are considered shallow and cold relative to the deep, warm snowpacks of the Sierra Nevada and Cascade mountains (Armstrong and Armstrong, 1987; Serreze et al., 1999; Trujillo and Molotch, 2014). We note in the discussion section of the original manuscript that these results are specific to the studied sites:

> "Firstly, we have only presented results from two sites within a single snow-dominated research catchment. Seasonal snow cover in the western United States spans a large elevational gradient and includes both maritime (e.g., the Cascades and Sierra Nevada) and continental (e.g., the Rocky Mountains) snowpack regimes (Serreze et al., 1999; Sturm et al., 1995). Therefore, an avenue for further research is to examine differences in cold content development across seasonally snow covered areas, with a particular focus on disentangling the effects of precipitation and air temperature during snowfall at sites with different snowpack characteristics."

However, we agree this framing overlooks the spatially extensive cold, shallow snowpacks of the Canadian Prairies and Arctic, and we have added text to the discussion mentioning these other snowpacks (p. 15 lines 17-21). We also reframed the last paragraph of the introduction (p. 3 lines 4-5) and research question 1 (p. 3 lines 9-10) per the recommendation of Reviewer 2 to emphasize that this research is specific to our study site.

Second, is that I'm not entirely convinced by the results. As I understand it, the authors assert via Figure 3 that cold content of the snow pack is explained by cumulative precipitation. A statistically significant trend line is show for the subalpine site; however, it has an r^2 of 0.17. Cold content is effectively an instantaneous, integrated snowpack temperature expressed as energy required to bring it to zero-degree isothermal. Cold content will, by definition, become greater (more negative) as below zero-degree mass is added to the snowpack. An r^2 of 0.17 is a poor correlation and does not, at least to me, act as strong evidence for the authors conclusion. Perhaps the r^2 for the alpine site is acceptable, however given cold content will by definition increase as cold mass is added, it seems to be a circular result that does not add any new knowledge nor should be unexpected.

We agree that an $r^2$ of 0.17 is low and we note this in the original manuscript:

> "The relationship was statistically significant at the 99% level at both sites despite the low coefficient of determination in the subalpine."

We use significant portions of the text to evaluate the differences between the alpine (precipitation explains the majority of the variance in cold content development and contributes over an order of magnitude more cold content than negative energy fluxes) and subalpine (precipitation explains a small portion of the variance but still contributes nearly an order of magnitude more cold content than negative energy fluxes). We have included additional text to reiterate the differences in between the alpine and subalpine (p. 8 lines 14-19; p. 9 lines 17-21).

Additionally, we frame in the introduction how little work has been done examining how cold content develops in seasonal snowpacks and that one of the only papers to do so suggests that it was primarily through a largely negative surface energy balance. Our conclusion is not that snowfall adds cold content (which is obvious and known), it is that snowfall is the dominant pathway through which the snowpacks at our two study sites develop cold content. This finding is interesting on its own in that the alpine site should have a high potential to develop cold content through a negative energy balance due to high rates of sublimation and net longwave emission from the snowpack. Surprisingly, despite this fact, precipitation exerts a stronger control on cold content in the alpine than subalpine.

With these results, the authors then proceed to the model step, effectively trying to duplicate the observed results. Stepping back, the message I feel like the authors are trying to present are: "there is no substantial radiative cooling of the snowpack, thus the precipitation temperature (and associated cold content) is the principal control on the total snowpack temperature, and therefore cold content." I suspect this is where Figure 8 becomes important, showing a small, negative total Qnet. However, something feels off about these results. In Figure 8a, the only real difference between day and night is the shortwave radiation and a slightly dampened latent heat flux. It seems odd to me that the mean response is identical, especially for the sensible heat flux. I'm just highly skeptical of an almost entirely similar surface energy balance between night and day. I would like the authors, upon confirming these results are correct as presented, to describe in more detail what is going on here, and if this is a site-specific effect or not, as my impression is it may be.

We thank Reviewer 1 for their critical evaluation of figure 8. This caused us to go back into the simulation data and take a closer look at why the values would be so similar, and we found another interesting facet of cold content development during non-snowfall days. When subsetting the data to all hours of cold content gain without snowfall, we found that midday gains (0900 h to 1400 h) were practically negligible. Thus, the energy balance results were similar for the day and night periods because the daytime gains occurred primarily in the early morning and late afternoon hours. We added Figure S1 (histograms showing the frequency of cold content gains without snowfall at each site, binned by hour) to the supplementary material and related text to the results section (p. 10 lines 4-11).

Because daytime hours with cold content gains were so uncommon, we decided to redo figure 8 to present the daily energy balance for non-snowfall days with cold content gains. Figure 8 now shows the energy balance for the entire day and mean $Q_{net}$ by hour. Interestingly, even on days with flux-driven cold content gains, $Q_{net}$ is positive during the midday hours. As noted in the paragraph above, midday hours with negative $Q_{net}$ were rare, thus the average $Q_{net}$ for these hours was positive.

Additionally, while taking a closer look at the data we found several instances of simulated cold content spiking up and down ($0.3 \text{ MJ m}^{-2} \text{ h}^{-1} \le \Delta CC \le -0.3 \text{ MJ m}^{-2} \text{ h}^{-1}$) before returning to approximately its

previous value. We removed these spikes from the analysis (representing less than 0.2% of the dataset) and added a note on p. 7 (lines 10-12).

Stepping back to Figure 6, I feel like this further highlights my issue with this conclusion. Full energy balance models use the balance of the energetics to simulate internal layer temperatures and energetics. Using cumulative mean air temperature feels very temperature-indexy and not really appropriate in this context – it supposes that the entirely of the snowpack energetics could potentially be explained by a mean air temperature, when in reality it's really the associated processes that would impact it.

We state our motivation for including air temperature in the introduction from the original manuscript:

> "Cold content can be estimated using at least one of three primary methods: 1) As an empirical function of air temperature (e.g., Anderson, 1976; Seligman et al., 2014; United States Army Corps of Engineers, 1956); 2) As a function of precipitation and air temperature (e.g., Cherkauer et al., 2003; Lehning et al., 2002b; Wigmosta et al., 1994) or wet bulb temperature (Anderson, 1968) during precipitation; and 3) As a residual of the snowpack energy balance (e,g., Andreadis et al., 2009; Cline, 1997; Lehning et al., 2002b; Marks and Winstral, 2001). In general, simple temperature-index models employ method 1, while both 2 and 3 are utilized in physics-based snow models. These methods suggest that cold content develops through both meteorological and energy balance processes, but few direct comparisons to observed cold content exist. This is likely due to the inherent difficulty in measuring cold content, which requires either time-intensive snow pits or co-located snow depth, density, and temperature measurements (Burns et al., 2014; Helgason and Pomeroy, 2011; Marks et al., 1992; Molotch et al., 2016). The lack of validation data introduces significant uncertainty into the dominant process by which cold content develops. Thus, it is not known whether cold content is primarily a function of air temperature (method 1), snowfall (method 2), or a negative surface energy balance (method 3)."

Given that air temperature is still used in current literature (e.g., DeWalle and Rango, 2008; Mosier et al., 2016; Seligman et al., 2014) to estimate cold content and that no research has shown the process (meteorological or energy balance) behind cold content development, we believe its inclusion appropriate.

Third, precipitation temperature and phase is unaddressed and is a critical component of this work. The simulations shown in Figure 9 c and d suppose the precipitation temperature and phase are correct. I'm assuming you used the default temperaturethreshold in Snowpack for phase? These results could be quite different if phase was wrong (i.e., rain instead of warm snow) or precipitation temperature was biased. There is substantial uncertainty associated with phase partitioning methods and snowfall temperature (e.g., Harder, et al. 2014), and these have significant implications for this work. How sensitive are these results to various phase and falling hydrometeor temperatures?

Regarding precipitation phase, we increased the standard SNOWPACK rain-snow air temperature threshold from 1.2°C to 2.5°C to better represent phase partitioning at our high-elevation continental location (a paper of ours in press at Nature Communications shows the Rocky Mountains have some of the highest rain-snow air temperature thresholds in the Northern Hemisphere; Jennings et al., In Press). To test the effect of our threshold selection, we compared the annual snow frequency using the 2.5°C threshold (alpine = 76.4%; subalpine = 61.5%) to a bivariate binary logistic regression phase prediction model (alpine = 76.7%; subalpine = 62.8%). This model predicts precipitation phase as a function of relative humidity and air temperature, and it was shown to the best precipitation phase method in a Northern Hemisphere comparison (Jennings et al., In Press). We have added this information to p. 6 (lines 10-16).

The temperature of precipitation is a likely shortcoming of SNOWPACK as the model sets precipitation temperature equal to air temperature. In independent work we have performed, we found wet bulb temperature to be a better predictor of new snow temperature, which was also noted in Harder and Pomeroy (2013). This should be included as an update in the SNOWPACK model considering wet bulb temperature could be easily estimated from the already standard forcing data. However, because SNOWPACK prescribes new snow temperature to be equal to air temperature, our estimates of the cold content added by precipitation are on the conservative side. The use of the colder wet bulb temperature (relative humidity is often below saturation, even during snowfall on Niwot Ridge) would lead to a greater amount of cold content added by precipitation. We have added this to the new modeling uncertainty discussion section (Sect. 5.2).

Fourth, despite reading through this a few times looking for it, it is unclear to me what kind of clearing this sub-alpine site is in. The site is specifically stated as a clearing, but the Snowpack canopy routine is enabled. This will significantly change the surface fluxes as well as precipitation at the snow surface; e.g., canopy interception. In my mind, this undermines the results presented herein – maybe it explains the poor result in Figure 3? – and needs to be detailed and the effects and impacts explained. Site photos would go a long way towards helping orient the reader. However as is, this is a major detail that is omitted.

This is an excellent point and we have added site photos to Figure 1. We have also changed the text from "small clearing" to "stand of lodgepole pine" to be clearer.

Fifth, A discussion on the role of Qg on cold content is needed and the assumptions behind your Qg simulation flux. These results show a treatment of the surface fluxes on cold content, but neglect discussion of soil-snowpack interactions, e.g., conditions that lead to frozen soil or refreezing of active layers.

We have added information to the manuscript discussing how $Q_G$ is simulated (covered in detail in Reviewer 1's other $Q_G$ comment on page 8 below). As to the rest of the comment, we do include $Q_G$ in our analysis of the snowpack energy balance, noting that it is typically positive even during periods of cold content gain. Frozen soil is more relevant to runoff processes and is outside the scope of this work. Refreezing of active layers leads to cold content losses (latent heat is released as the phase of water changes from liquid to solid). While this process is important to snowpack ripening and snowmelt generation, we only consider the empirical relationships between cold content and snowmelt rate/timing in this work.

Lastly, the authors assert that increased peak cold content and total spring precipitation control snowmelt onset. But this seems by-definition – doesn't this imply more mass and refreshed albedos? Isn't this just what you'd expect with increased cold content being a function of snowpack mass?

This is noted in the discussion of the original manuscript:

> "These results all suggest later seasonal snowmelt onset and faster snowmelt rates are primarily a function of persistent snowfall. While snowfall events can add significant cold content to the snowpack, they also change other fundamental properties that can delay snowmelt timing, such as increasing surface albedo (Clow et al., 2016) and adding dry pore space that must be saturated (Seligman et al., 2014)."

In summary: As I understand the results presented, the story is that the authors found limited evidence for sustained energy loss from the snowpack and that the cold content of the snowpack was mostly a result of mass inputs. However, there are many confounding factors that make it difficult to accept this at face

value. Given the circular reasoning in the results (more snow -> more cold content, but that is by definition), it is difficult for the reader to accept the results. That being said, validating the model against these observations is quite interesting and diagnosing snowpack energy loss during the winter is a useful contribution. However, I think the overall message needs to be refined to more clearly articulate the site-specific nature of this study, the uncertainties in key aspects of the analysis (e.g., precipitation, canopy), and the text improved for readability.

We again thank Reviewer 1 for their critical review of this work. We have added text to the manuscript to illustrate our results are specific to our two study sites in addition to the discussion section that covered this point in the original manuscript. We have also addressed their specific comments below to improve the readability of the text and more clearly outline the project's uncertainties.

Additionally, we would like to reiterate the novelty of this work. There is relatively little previous literature assessing the meteorological and energy balance controls on cold content development. We used a combination of observed data and validated simulation output to show precipitation was the dominant source of cold content development at our two sites. This finding was particularly surprising at the cold alpine site considering its high rates of snowpack sublimation and net longwave emission.

References Harder, P., and J. W. Pomeroy (2014), Hydrological model uncertainty due to precipitation-phase partitioning methods, Hydrol. Process., 28, 4311–4327, doi:10.1002/hyp.10214.

Specific points

Throughout:
The authors introduce (para 25) increase/decrease for cold content, but proceed to use gain/loss. I think it should be consistent throughout

We have changed p. 5 (lines 4-6) to include gain/loss and increase/decrease.

Figure is used in the text but Fig. when used in brackets. Ideally should be consistent.

This usage is in accordance with The Cryosphere's style guide (https://www.the-cryosphere.net/for_authors/manuscript_preparation.html) and will be kept:

> *"The abbreviation "Fig." should be used when it appears in running text and should be followed by a number unless it comes at the beginning of a sentence, e.g.: "The results are depicted in Fig. 5. Figure 9 reveals that...".*

Units should be separated with a cdot instead of spaces, e.g., Wâ´NEˇmˆ(-2)

The Cryosphere does not specify the use of c-dots. The only example units we found in the manuscript prep instructions showed the units with a space as we have in the manuscript:

> *"Units must be written exponentially (e.g. W m$^{-2}$)."*

Unclear what wet and dry days mean. Wet implies rain to me, but I suspect that's not what you mean. I would reword, or at least clearly define.

Yes, this is confusing. We have changed all relevant text to reflect this (snowfall/precipitation and non-snowfall/non-precipitation days instead of wet and dry days).

P1, Para 20: "cold content ... associated with reduced snowmelt" this needs to be reworded as snowmelt should be happening when CC = 0. Which melt rate is being considered?

Reworded for clarity.

P2, Para 20: "the authors" which authors?

Reworded for clarity.

P2, Para 25: "Furthermore: : :" I'm not sure I agree with this statement. CC needs to be = 0 for melt to occur, so isn't this known? Do you have a citation?

The citations are in the following paragraphs and they indicate they provide differing perspectives on the control exerted by cold content on seasonal melt rate and timing. However, our text was not clear enough (yes, melt occurs when $Q_{net}$ is positive and CC = 0) that we were referring to winter cold content magnitude and we have edited p. 2 (line 29) for clarity.

P2, Para 30: "saturate", word choice

Saturate is commonly used in the snow hydrology literature in reference to satisfying the irreducible liquid water content of a snowpack, but we have edited p. 3 (lines 1-2) to be more specific and be applicable to other hydrologists to whom the word saturate may indicate all pore space is filled.

P2, Para 30: "However: : :", I'm unclear what you're trying to say, please clarify.

Edited for clarity.

P3, Para 10, 15, 30 Need to be indented.

Changed

P3, Para 20, use "10 m/s to 13 m/s" instead of how it is written.

Changed, but units are left in exponential form to be consistent with The Cryosphere style.

P4, Para 1, "Snow" incorrect capitalized

Snow begins an independent clause, meaning it can (or should, depending on your preferred style guide) be left capitalized.

P4, Para 14, "downwelling longwave" I would put a quick note as to what method you used.

Added this information to p.4 (lines 21-22), with full methodology detailed in the appendix.

P5, Para 20, remove "proposed in Sect. 1"

Changed

P5, Para 20 "We then quantified" I found this section unclear

Rewritten for clarity.

P5, Eqn 3 Consider writing 86,400 as a variable and showing in the text the units. Either way, you need units.

Changed.

P5, Para 15 "in order to improve" Using a model doesn't improve obs, it just compliments them. I think you should reword to make this distinction.

Yes, changed.

P5, Para 20 "number of finite elements" change to layers

Changed. Although SNOWPACK is a finite element model, that is not important here.

P5, Para 25 remove "the numerical model in"

Removed.

P5, Para 5, the canopy module stuff comes out of nowhere, especially given you say the site is in a clearing. This needs to be much clearer.

We changed the site description to note the pits are dug in a "stand" of lodgepole pine and have included a site photo in Figure 1.

P5, Para 20 "Output from snow model simulations" I don't follow. Do you mean the comparison is more robust w/multiple outputs to validate?

We are noting that the output of snow model simulations has greater fidelity when validated on more than just SWE, based on the work of Lapo et al. (2015). We are stating that we can make conclusions on the simulations of snowpack cold content because we have actual measurements of cold content to which we can compare the model output. We added text to p. 7 (lines 4-5) to clarify.

P6, Para 20 Any EC observations considered?

We did not consider using the EC observations on Niwot Ridge because the subalpine AmeriFlux tower measures fluxes above the canopy (21 m) and not near the snow surface. There are EC measurements in the alpine, but the records are short and the instruments are located near areas of snow scour.

P7, Eqn 4 The form for the energy balance equation given in Equation 4 is not a standard form. Generally, the change in internal energetics are given as a dU/dt and Qm is on the LHS. Qnet and Qm together are redundant in the energy balance as the energy available for melt is the net energy.

The form presented in the Equation 4 is used frequently throughout the snow hydrology literature (Cline, 1997; Marks et al., 2008; Marks and Dozier, 1992, to name just a few examples). However, we have changed the notation to the suggested form in order to avoid confusion. Additionally, we have changed $Q_{net}$ to refer to the sum of the radiative, turbulent, and ground heat fluxes in order to avoid writing out the full energy balance each time we are referring to the net surface and ground fluxes.

P7, Para 1, "time scales" -> temporal scales

Time scale is appropriate and left as-is.

P7, Para 25, as I said above, I don't buy that an r^2=0.17 demonstrates a primary control

Please see our previous note. We also clarified to say "of the two meteorological quantities evaluated here" to note we are referring to precipitation and air temperature. As mentioned previously, we devote an entire discussion section to the topic of subalpine cold content development and why precipitation exhibits a reduced effect relative to the alpine.

P8, Para 5, Probably should note these are depth averaged

Fixed.

P8, Para 10, -2.2 should have units after it P8,

Fixed.

Para 15, How is this working with the canopy module? Intercepted snow has massive sublimation losses, but that doesn't seem to be reflected here.

We are only concerned with snow surface sublimation as canopy sublimation does not directly lead to changes in snowpack internal energy.

P8, Para 20, Monotonically is either monotonic or not. There is no in-between. Reword

Correct. We removed that sentence and included a new sentence to clarify precipitation exerts a stronger control in the alpine than subalpine (p. 9, lines 17–21).

P8, Para 25, "simulations confirm" change to "support" or similar

Changed.

P9, Para 10, So how are you calculating Qg? Maybe I missed it? I think you need a reasonable treatment on the assumptions behind however you do this. Did you couple snowpack with the soil? Constant flux? Constant ground temp? Qg is important for a conduction heat flux into the snow pack, and needs to be addressed if you go after cold content. Often Qg is taken to be 0-4W/m^2, but this flux can be important for stopping a numerical model from simulating absurd cold contents.

We used version 3.3 of SNOWPACK, which assumes a ground temperature of 0°C when there is snow cover. Simulations showed $Q_G$ was typically between 0 and 4 W m$^{-2}$ when snow depth exceeded ~20 cm, which is similar to other values reported in the literature. Thus, $Q_G$ provides a small, but consistently positive to the snowpack energy balance. Cline (1997) noted ground heat flux was negligible at the alpine site using a flux plate in the soil. Snow pit data from the alpine and subalpine are consistent with this in that the warmest snowpack temperatures are observed at the bottom until ripening begins. We added this information to p. 6 (lines 20-21) in the methods and new Sect. 5.2 in the discussion.

P12, Para 10 "continued snowfall" But this is just more mass, so you'd expect snowmelt timing to be delayed P12,

Mass additions do not lead to consistent snowmelt responses because mass in and of itself is not a physical property that delays snowmelt (i.e., a given amount of warm, wet snow has a much different

effect on snow energetics and runoff than the same amount of cold, dry snow due to the amount of liquid water vs. dry pore space, changes to surface albedo, and cold content). Furthermore, we note other hypotheses in the discussion of the original manuscript:

> "These results all suggest later seasonal snowmelt onset and faster snowmelt rates are primarily a function of persistent snowfall. While snowfall events can add significant cold content to the snowpack, they also change other fundamental properties that can delay snowmelt timing, such as increasing surface albedo (Clow et al., 2016) and adding dry pore space that must be saturated (Seligman et al., 2014)."

Para 20, "future work: : :" Lots of work on this already: : :.

We have rewritten to clarify that most previous work focuses on single, well-instrumented sites (such as our manuscript) or vast networks of SNOTEL-like sites with only air temperature, precipitation, and SWE observations (e.g., Trujillo and Molotch, 2014). A spatially explicit, energy balance treatment has yet to be applied to the different snow categories of the western United States despite the irreplaceable contribution of snowmelt to the hydrologic cycle and regional water resources. We have added text to this discussion section to clarify our statement (p. 14 lines 29-32).

P16, Para 30 Given Snowpack is forced hourly, this longwave estimate seems like a massive source of uncertainty, especially within the context of an energy balance model. There are many incoming longwave formulations that take into account various proxies for non-clear sky. You seem to do this for your emissivity, but it's not clear how that exactly works. With such low r^2 this needs to be detailed and expanded upon. The large error in a critically important mid-winter energy flux may have substantial implications for this work.

This is an important point and we have added a note on this limitation to the new discussion section on modeling uncertainty (Sect. 5.2). Downwelling longwave radiation is under-sampled relative to other standard forcing data (Raleigh et al., 2016) and associated errors propagate into SWE and snow temperature biases (Lapo et al., 2015). Schlögl et al. (2016) showed little sensitivity in Alpine3D SWE simulations to a selection of empirical longwave estimates and Lapo et al. (2015) indicated the largest effects of longwave uncertainty were simulated at perturbations greater than $\pm 10$ W m$^{-2}$. Thus our small mean bias likely indicates the total amount of incoming longwave radiation is correct while the low $r^2$ suggests the timing of subdaily fluctuations is not well simulated. Our multi-variable validation shows that SNOWPACK performs well relative to snow pit observations of SWE, depth-weighted snowpack temperature, and cold content.

Figures All figures – It would certainly aid readability to have them labeled as alpine/sub alpine without having to constantly refer to the caption.

Agreed. Changed on all relevant figures.

Figure 1, difficult to determine differences at high elevation.

We added contour lines to the figure along with site photos per an earlier recommendation.

Figure 2, can you change the DOY to dates for easier parsing?

Yes, changed for easier interpretation on this figure and others that previously showed DOWY.

Figure 5a,b Should have same axis extents

Axes left as-is.

Figure 8abcd would benefit from having the same y- (ab) and x- (cd) axes to aid in comparison. Also, please expand the y-axes of (ab) so-as to understand what the limits are.

Changed per this suggestion and our notes on the energy balance in pages 2 and 3 above.

Figure 9, needs legend

Changed.

**References**
Anderson, E.A., 1976. A point of energy and mass balance model of snow cover. NOAA Tech Rep NWS 19, 1–150.
Anderson, E.A., 1968. Development and testing of snow pack energy balance equations. Water Resour. Res. 4, 19–37.
Andreadis, K.M., Storck, P., Lettenmaier, D.P., 2009. Modeling snow accumulation and ablation processes in forested environments. Water Resour. Res. 45. https://doi.org/10.1029/2008WR007042
Armstrong, R.L., Armstrong, B.R., 1987. Snow and avalanche climates of the western United States: a comparison of maritime, intermountain and continental conditions. IAHS Publ 162, 281–294.
Burns, S.P., Molotch, N.P., Williams, M.W., Knowles, J.F., Seok, B., Monson, R.K., Turnipseed, A.A., Blanken, P.D., 2014. Snow Temperature Changes within a Seasonal Snowpack and Their Relationship to Turbulent Fluxes of Sensible and Latent Heat. J. Hydrometeorol. 15, 117–142. https://doi.org/10.1175/JHM-D-13-026.1
Cherkauer, K.A., Bowling, L.C., Lettenmaier, D.P., 2003. Variable infiltration capacity cold land process model updates. Glob. Planet. Change 38, 151–159. https://doi.org/10.1016/S0921-8181(03)00025-0
Cline, D.W., 1997. Snow surface energy exchanges and snowmelt at a continental, midlatitude Alpine site. Water Resour. Res. 33, 689–701.
Clow, D.W., Williams, M.W., Schuster, P.F., 2016. Increasing aeolian dust deposition to snowpacks in the Rocky Mountains inferred from snowpack, wet deposition, and aerosol chemistry. Atmos. Environ., Acid Rain and its Environmental Effects: Recent Scientific AdvancesPapers from the Ninth International Conference on Acid Deposition 146, 183–194. https://doi.org/10.1016/j.atmosenv.2016.06.076
DeWalle, D.R., Rango, A., 2008. Principles of snow hydrology. Cambridge University Press.
Harder, P., Pomeroy, J., 2013. Estimating precipitation phase using a psychrometric energy balance method. Hydrol. Process. 27, 1901–1914. https://doi.org/10.1002/hyp.9799
Helgason, W., Pomeroy, J., 2011. Problems Closing the Energy Balance over a Homogeneous Snow Cover during Midwinter. J. Hydrometeorol. 13, 557–572. https://doi.org/10.1175/JHM-D-11-0135.1
Jennings, K.S., Winchell, T.S., Livneh, B., Molotch, N.P., In Review. Spatial variation of the rain-snow temperature threshold across the Northern Hemisphere. Nat. Commun.

Lapo, K.E., Hinkelman, L.M., Raleigh, M.S., Lundquist, J.D., 2015. Impact of errors in the downwelling irradiances on simulations of snow water equivalent, snow surface temperature, and the snow energy balance. Water Resour. Res. 51, 1649–1670.

Lehning, M., Bartelt, P., Brown, B., Fierz, C., 2002. A physical SNOWPACK model for the Swiss avalanche warning: Part III: Meteorological forcing, thin layer formation and evaluation. Cold Reg. Sci. Technol. 35, 169–184.

Marks, D., Dozier, J., 1992. Climate and energy exchange at the snow surface in the alpine region of the Sierra Nevada: 2. Snow cover energy balance. Water Resour. Res. 28, 3043–3054.

Marks, D., Dozier, J., Davis, R.E., 1992. Climate and Energy Exchange at the Snow Surface in the Alpine Region of the Sierra Nevada 1. Meteorological Measurements and Monitoring. Water Resour. Res. 28, 3029–3042.

Marks, D., Winstral, A., 2001. Comparison of snow deposition, the snow cover energy balance, and snowmelt at two sites in a semiarid mountain basin. J. Hydrometeorol. 2, 213–227.

Marks, D., Winstral, A., Flerchinger, G., Reba, M., Pomeroy, J., Link, T., Elder, K., 2008. Comparing simulated and measured sensible and latent heat fluxes over snow under a pine canopy to improve an energy balance snowmelt model. J. Hydrometeorol. 9, 1506–1522.

Molotch, N.P., Barnard, D.M., Burns, S.P., Painter, T.H., 2016. Measuring spatiotemporal variation in snow optical grain size under a subalpine forest canopy using contact spectroscopy. Water Resour. Res. https://doi.org/10.1002/2016WR018954

Mosier, T.M., Hill, D.F., Sharp, K.V., 2016. How much cryosphere model complexity is just right? Exploration using the conceptual cryosphere hydrology framework. The Cryosphere 10, 2147–2171. https://doi.org/10.5194/tc-10-2147-2016

Raleigh, M.S., Livneh, B., Lapo, K., Lundquist, J.D., 2016. How Does Availability of Meteorological Forcing Data Impact Physically Based Snowpack Simulations? J. Hydrometeorol. 17, 99–120. https://doi.org/10.1175/JHM-D-14-0235.1

Schlögl, S., Marty, C., Bavay, M., Lehning, M., 2016. Sensitivity of Alpine3D modeled snow cover to modifications in DEM resolution, station coverage and meteorological input quantities. Environ. Model. Softw. 83, 387–396. https://doi.org/10.1016/j.envsoft.2016.02.017

Seligman, Z.M., Harper, J.T., Maneta, M.P., 2014. Changes to Snowpack Energy State from Spring Storm Events, Columbia River Headwaters, Montana. J. Hydrometeorol. 15, 159–170. https://doi.org/10.1175/JHM-D-12-078.1

Serreze, M.C., Clark, M.P., Armstrong, R.L., McGinnis, D.A., Pulwarty, R.S., 1999. Characteristics of the western United States snowpack from snowpack telemetry (SNOTEL) data. Water Resour. Res. 35, 2145–2160.

Sturm, M., Holmgren, J., Liston, G.E., 1995. A seasonal snow cover classification system for local to global applications. J. Clim. 8, 1261–1283.

Trujillo, E., Molotch, N.P., 2014. Snowpack regimes of the Western United States. Water Resour. Res. 50, 5611–5623. https://doi.org/10.1002/2013WR014753

United States Army Corps of Engineers, 1956. Snow hydrology. US Army North Pac. Div. Portland Or.

Wigmosta, M.S., Vail, L.W., Lettenmaier, D.P., 1994. A distributed hydrology-vegetation model for complex terrain. Water Resour. Res. 30, 1665–1679.

---

## Author Comment (AC2) · 7 Mar 2018

**Review of :**

**Observations and simulations of the seasonal evolution of snowpack cold content and its relation to snowmelt and the snowpack energy budget,
by Jennings et al.**

The authors address the issue of the drivers of cold content evolution based on the exemple of two seasonal snowpacks from a single observation catchment in the Western US. They also assess, in a much shorter part, the effect of cold-content on snow-melt timing and rates.

The paper is well written and well illustrated. The take-home message is clear and the objectives assessed in Introduction are achieved with scientific quality. I found the paper both an appreciable synthesis of existing litterature on the topic, and enlightening regarding the conclusions achieved.

We thank Reviewer 2 for their suggestions and thoughtful review of the manuscript. Our responses are in blue text throughout this document and we have made changes to the manuscript in regard to their comments.

In addition to the few suggestions made below, there is in my opinion one minor contradiction in the Result section (point 9 below), that I would recommand the authors to adress with priority as it affects the consistency of the paper. Note that this apparent contradiction may come from misunderstanding from my side, or an edit mistake from the side of the authors. Comment addressed in point 9 below.

1.  Introduction-p2 Ll-6 : not the snowmelt itself is critical for the cited applications, but the timing of surface/subsurface runoff from Snowmelt, which may not be the same when surface melt occurs and refreezes (as just mentionned earlier in the manuscript). For consistence I suggest changing« snowmelt » into« runoff from snowmelt » in the current sentence.
    Yes, this is a good distinction. We have changed the text to reflect this (p. 2 lines 2-3).

2.  Introduction-p2 L16-18 : Aren't the « dominant processes »well-resolved by Method-3 (residual of the energy balance) at places where energy-balance models like SNOWPACK are routinely validated against data regarding most components of the energy balance ? In that case, isn't it rather a lack of investigation into the process of cold content development, than a lack of validation data, that limits knowledge of the prevailing processes ?
    Given the adequate performance of advanced energy balance models (Etchevers et al., 2004; Rutter et al., 2009), it is somewhat reasonable to assume they are accurately simulating the evolution of snowpack cold content. However, recent work has shown the utility in validating snow model output on more than one state variable (Lapo et al., 2015) in order to ensure we are not "getting the right answers for the wrong reasons." In this case we stand by our assertion that having measurements of snowpack temperature and cold content are necessary for making conclusions on how cold content develops in seasonal snowpacks. I.e., previous work could have tested hypotheses using energy balance model output alone, but a lack of cold content validation data would have limited the strength of their conclusions.

    Furthermore, the dominant processes involved in cold content development likely depend on the climatology of the investigated sites. Event though it is well stated in the Discussion, specifying the perimeter of validity of your study should be done here already. Typically, I suggest transforming research question #1(p3 L6) into « What are the meteorological and energy balance controls on cold content development **at two alpine and sub-alpine sites from the Western US ? »**

Yes, we agree with this point and have changed research question 1 to reflect Reviewer 2's suggestion (p. 3 lines 9-10). This point was also noted by Reviewer 1, so we have added text wherever possible to note the results are specific to our study sites.

3.  Introduction-p2 L22-26 : conduction fluxes within the snowpack, and from snowpack to the ground, can mitigate the impact of the intense negative fluxes reported here. If a gradient around 100 W/m develops within the snowpack as a result of intense surface cooling, around 20 W/m2 propagates downwards (upon hypothesis of a 0.2 W/m/K conductivity for Snow), which should somewhat prevents the snowpack from locally reaching unreaslitic temperatures ( ?)

    In the cited work, the authors note that their reported values are for ∆Q, or the change in snowpack internal energy, (Marks and Dozier, 1992). Although their paper has provided many significant contributions to the state of knowledge of the snowpack energy balance, it also underlines how little we previously knew about cold content development processes. We hope our manuscript provides a small step in the right direction.

4.  Introduction-p3 Lĺ : uncertainties->unknowns(suggestion)

    Changed.

5.  Methods - p5 L28 : « vapour diffusion » in SNOWPACK is actually only calculated to compute Snow grain/bounds growth rates. There is no mass redistribution between different snow layers as a result of vapour diffusion in current versions of SNOWPACK. I therefore suggest to suppress this item from the list of existing SNOWPACK routines, as it would be otherwise misleading.

    Thank you for clarifying. Given this paper is not about grain metamorphism, we have removed this part.

6.  Methods- p6 L7-10 : could you specify here or in appendix the result of your calibration procedure for the parameters leaf area index, vegetation height, direct canopy throughfall, and wind speed reduction ? Note that these parameters are usually estimated from field data, and that any observation-based estimate of them would help assess the soundness of the calibrated parameter or of the canopy model.

    Yes. We have added this information to the Supplemental Material (Table S2).

    Additionnal, the rough size of the clearing where sub-alpine snowpits were made, should be specified (p4Lĺ) to justify the use the canopy module of SNOWPACK, instead of an open- area version SNOWPACK with just wind attenuation.

    We have included site photos for both locations (Figure 1) in the updated manuscript and removed the "small clearing" part from the text as this caused some confusion.

7.  Results-p5 L15-16 : « Peak cold content and peak SWE respectively occurred 33 d and 10 d later in the alpine than subalpine ».Add « on average » to this sentence and the next.

    Changed.

8.  Results-p7L25-27 : « This is likely due to the increased variability of winter precipitation, the coefficient of variation of which is 2.9 and 2.7 times greater than that of air temperature in the alpine and subalpine, respectively ». I assume that by « increased »you mean« higher» ? I would suggest that snow-atmosphere heat transfers occuring during cold air temperatures periods are less efficient in cooling the

snowpack, than the direct addition of cold Snow from fresh snowfall.

Correct, we have changed *increased* to *higher*. Additionally, we have added text to Sect. 4.1 to note that the air temperature is less effective at producing cold content than precipitation.

9. Results- p7L29-30 : « During periods of SWE accumulation, $Q_{net}$ was typically near 0 W m$^{-2}$ (Fig. 4a), indicating a large negative energy balance was not responsible for cold content development. » First, here, you infer $Q_{net}$ from the variation in CC between 2 snowpit dates, so where is the link to energy balance ? Second, based on Eq 3, $Q_{net}$ ~0 W m$^{-2}$ indicates no cold-content increase, meaning there is no visible snowfall-driven cold-content increase in the snowpit data. In my mind this contradicts the other results of the study, e.g. Fig 3 and 7 - please justify, or explain me where I am wrong. May I suggest using different names for $Q_{net}$ in Eq. 3 and $Q_{net}$ in Eq. 4 ? Like $Q_{net-pit}$ and $Q_{net-EB}$ respectively.

This is a good point and similar to the one Reviewer 1 brought up regarding our energy balance notation. In the quoted lines we were attempting to convey that the snow pits showed no direct evidence of a large negative surface energy balance like the one reported in Marks and Dozier (1992). We computed $Q_{net}$ as a function of the change in cold content and the time between pit observations. To be more consistent, and clearer, we have changed our notation throughout the paper to have dU/dt be the change in internal energy of the snowpack (dU/dt$_{pit}$ for the snow pit data) and $Q_{net}$ to represent the sum of SW$_{net}$, LW$_{net}$, $Q_H$, $Q_{LE}$, and $Q_G$ (per the recommendation of Reviewer 1).

In regard to your point "$Q_{net}$ ~0 W m$^{-2}$ indicates no cold-content increase, meaning there is no visible snowfall-driven cold-content increase in the snowpit data", we have clarified in lines 20-27 (p. 8) the way we presented these data. Because cold content is a relatively small value in terms of W m$^{-2}$, decreases in dU/dt will always be small, whether cold content gains come through precipitation or a negative surface energy balance. For example, if two snow pits were dug exactly one day apart, the computed dU/dt for a 0.2 MJ m$^{-2}$ increase in cold content (2 cm of new SWE at -5°C) would be just -2.4 W m$^{-2}$ (assuming all other energy balance components summed to zero). Thus, a fairly significant cold snowfall event would show up as a very small dU/dt value.

10. Results- p8 L7 : could overestimated densities be the reason for cold-content overestimation at alpine location ? (as Snow temperature tend to be overestimated ?) Maybe a line on that could be added to the Result or Discussion section

We have added this point in the results section (p. 9 lines 3-6) and we have also added a new discussion section on the model shortcomings (Sect. 5.2).

**References**

Etchevers, P., Martin, E., Brown, R., Fierz, C., Lejeune, Y., Bazile, E., Boone, A., Dai, Y.-J., Essery, R., Fernandez, A., others, 2004. Validation of the energy budget of an alpine snowpack simulated by several snow models (SnowMIP project). Ann. Glaciol. 38, 150–158.

Lapo, K.E., Hinkelman, L.M., Raleigh, M.S., Lundquist, J.D., 2015. Impact of errors in the downwelling irradiances on simulations of snow water equivalent, snow surface temperature, and the snow energy balance. Water Resour. Res. 51, 1649–1670.

Marks, D., Dozier, J., 1992. Climate and energy exchange at the snow surface in the alpine region of the Sierra Nevada: 2. Snow cover energy balance. Water Resour. Res. 28, 3043–3054.

Rutter, N., Essery, R., Pomeroy, J., Altimir, N., Andreadis, K., Baker, I., Barr, A., Bartlett, P., Boone, A., Deng, H., others, 2009. Evaluation of forest snow processes models (SnowMIP2). J. Geophys. Res. Atmospheres 114.

---

## Author Response (AR1)

The authors present a study that uses a long term observational data set to validate simulated snowpack cold content. The authors attribute the largest increase in cold content to new precipitation mass. Validating a complex, multi-layer snowpack model that is frequently used in the literature is a substantial contribution, especially given the uniqueness of the long-term snow pit data. However, as currently presented, this manuscript needs substantial revision and polish. Below I explain my reasoning for this, and I hope the authors can use it to improve this manuscript into the contribution that is hiding under the surface. As is, I recommend accept pending major revisions.

Thank you for the detailed, critical review and the suggestion to publish pending the revisions. In the below response, our comments are in blue text.

My first issue is that these conclusions are specific to a deep snowpacks in a warmer climate. Thin, shallow snowcovers have a long record in the literature as being difficult to simulate due to the substantial radiative cooling of the snowpack resulting in sharp gradients and maximum cold content being exceeded. It is important that all these results are very clearly stated to apply to the deep snowpacks herein.

Given our geographic setting (Colorado Rocky Mountains), we framed this work in the context of western US snowpacks, which are essential to regional water resources. In this regard, the alpine and subalpine snowpacks are considered shallow and cold relative to the deep, warm snowpacks of the Sierra Nevada and Cascade mountains (Armstrong and Armstrong, 1987; Serreze et al., 1999; Trujillo and Molotch, 2014). We note in the discussion section of the original manuscript that these results are specific to the studied sites:

"Firstly, we have only presented results from two sites within a single snow-dominated research catchment. Seasonal snow cover in the western United States spans a large elevational gradient and includes both maritime (e.g., the Cascades and Sierra Nevada) and continental (e.g., the Rocky Mountains) snowpack regimes (Serreze et al., 1999; Sturm et al., 1995). Therefore, an avenue for further research is to examine differences in cold content development across seasonally snow covered areas, with a particular focus on disentangling the effects of precipitation and air temperature during snowfall at sites with different snowpack characteristics."

However, we agree this framing overlooks the spatially extensive cold, shallow snowpacks of the Canadian Prairies and Arctic, and we have added text to the discussion mentioning these other snowpacks (p. 15 lines 17-21). We also reframed the last paragraph of the introduction (p. 3 lines 4-5) and research question 1 (p. 3 lines 9-10) per the recommendation of Reviewer 2 to emphasize that this research is specific to our study site.

Second, is that I'm not entirely convinced by the results. As I understand it, the authors assert via Figure 3 that cold content of the snow pack is explained by cumulative precipitation. A statistically significant trend line is show for the subalpine site; however, it has an r2 of 0.17. Cold content is effectively an instantaneous, integrated snowpack temperature expressed as energy required to bring it to zero-degree isothermal. Cold content will, by definition, become greater (more negative) as below zero-degree mass is added to the snowpack. An r2 of 0.17 is a poor correlation and does not, at least to me, act as strong evidence for the authors conclusion. Perhaps the r2 for the alpine site is acceptable, however given cold content will by definition increase as cold mass is added, it seems to be a circular result that does not add any new knowledge nor should be unexpected.

We agree that an  $r^2$  of 0.17 is low and we note this in the original manuscript:

"The relationship was statistically significant at the 99% level at both sites despite the low coefficient of determination in the subalpine."

We use significant portions of the text to evaluate the differences between the alpine (precipitation explains the majority of the variance in cold content development and contributes over an order of magnitude more cold content than negative energy fluxes) and subalpine (precipitation explains a small portion of the variance but still contributes nearly an order of magnitude more cold content than negative energy fluxes). We have included additional text to reiterate the differences in between the alpine and subalpine (p. 8 lines 14-19; p. 9 lines 17-21).

Additionally, we frame in the introduction how little work has been done examining how cold content develops in seasonal snowpacks and that one of the only papers to do so suggests that it was primarily through a largely negative surface energy balance. Our conclusion is not that snowfall adds cold content (which is obvious and known), it is that snowfall is the dominant pathway through which the snowpacks at our two study sites develop cold content. This finding is interesting on its own in that the alpine site should have a high potential to develop cold content through a negative energy balance due to high rates of sublimation and net longwave emission from the snowpack. Surprisingly, despite this fact, precipitation exerts a stronger control on cold content in the alpine than subalpine.

With these results, the authors then proceed to the model step, effectively trying to duplicate the observed results. Stepping back, the message I feel like the authors are trying to present are: "there is no substantial radiative cooling of the snowpack, thus the precipitation temperature (and associated cold content) is the principal control on the total snowpack temperature, and therefore cold content." I suspect this is where Figure 8 becomes important, showing a small, negative total Qnet. However, something feels off about these results. In Figure 8a, the only real difference between day and night is the shortwave radiation and a slightly dampened latent heat flux. It seems odd to me that the mean response is identical, especially for the sensible heat flux. I'm just highly skeptical of an almost entirely similar surface energy balance between night and day. I would like the authors, upon confirming these results are correct as presented, to describe in more detail what is going on here, and if this is a site-specific effect or not, as my impression is it may be.

We thank Reviewer 1 for their critical evaluation of figure 8. This caused us to go back into the simulation data and take a closer look at why the values would be so similar, and we found another interesting facet of cold content development during non-snowfall days. When subsetting the data to all hours of cold content gain without snowfall, we found that midday gains (0900 h to 1400 h) were practically negligible. Thus, the energy balance results were similar for the day and night periods because the daytime gains occurred primarily in the early morning and late afternoon hours. We added Figure S1 (histograms showing the frequency of cold content gains without snowfall at each site, binned by hour) to the supplementary material and related text to the results section (p. 10 lines 4-11).

Because daytime hours with cold content gains were so uncommon, we decided to redo figure 8 to present the daily energy balance for non-snowfall days with cold content gains. Figure 8 now shows the energy balance for the entire day and mean  $Q_{net}$  by hour. Interestingly, even on days with flux-driven cold content gains,  $Q_{net}$  is positive during the midday hours. As noted in the paragraph above, midday hours with negative  $Q_{net}$  were rare, thus the average  $Q_{net}$  for these hours was positive.

Additionally, while taking a closer look at the data we found several instances of simulated cold content spiking up and down (0.3 MJ m-2 h-1  $\leq \Delta CC \leq -0.3$  MJ m-2 h-1) before returning to approximately its

previous value. We removed these spikes from the analysis (representing less than 0.2% of the dataset) and added a note on p. 7 (lines 10-12).

Stepping back to Figure 6, I feel like this further highlights my issue with this conclusion. Full energy balance models use the balance of the energetics to simulate internal layer temperatures and energetics. Using cumulative mean air temperature feels very temperature-indexy and not really appropriate in this context – it supposes that the entirely of the snowpack energetics could potentially be explained by a mean air temperature, when in reality it's really the associated processes that would impact it.

**We state our motivation for including air temperature in the introduction from the original manuscript:**

"Cold content can be estimated using at least one of three primary methods: 1) As an empirical function of air temperature (e.g., Anderson, 1976; Seligman et al., 2014; United States Army Corps of Engineers, 1956); 2) As a function of precipitation and air temperature (e.g., Cherkauer et al., 2003; Lehning et al., 2002b; Wigmosta et al., 1994) or wet bulb temperature (Anderson, 1968) during precipitation; and 3) As a residual of the snowpack energy balance (e.g., Andreadis et al., 2009; Cline, 1997; Lehning et al., 2002b; Marks and Winstral, 2001). In general, simple temperature-index models employ method 1, while both 2 and 3 are utilized in physics-based snow models. These methods suggest that cold content develops through both meteorological and energy balance processes, but few direct comparisons to observed cold content exist. This is likely due to the inherent difficulty in measuring cold content, which requires either time-intensive snow pits or co-located snow depth, density, and temperature measurements (Burns et al., 2014; Helgason and Pomeroy, 2011; Marks et al., 1992; Molotch et al., 2016). The lack of validation data introduces significant uncertainty into the dominant process by which cold content develops. Thus, it is not known whether cold content is primarily a function of air temperature (method 1), snowfall (method 2), or a negative surface energy balance (method 3)."

Given that air temperature is still used in current literature (e.g., DeWalle and Rango, 2008; Mosier et al., 2016; Seligman et al., 2014) to estimate cold content and that no research has shown the process (meteorological or energy balance) behind cold content development, we believe its inclusion appropriate.

Third, precipitation temperature and phase is unaddressed and is a critical component of this work. The simulations shown in Figure 9 c and d suppose the precipitation temperature and phase are correct. I'm assuming you used the default temperaturethreshold in Snowpack for phase? These results could be quite different if phase was wrong (i.e., rain instead of warm snow) or precipitation temperature was biased. There is substantial uncertainty associated with phase partitioning methods and snowfall temperature (e.g., Harder, et al. 2014), and these have significant implications for this work. How sensitive are these results to various phase and falling hydrometeor temperatures?

Regarding precipitation phase, we increased the standard SNOWPACK rain-snow air temperature threshold from  $1.2^{\circ}$ C to  $2.5^{\circ}$ C to better represent phase partitioning at our high-elevation continental location (a paper of ours in press at Nature Communications shows the Rocky Mountains have some of the highest rain-snow air temperature thresholds in the Northern Hemisphere; Jennings et al., In Press). To test the effect of our threshold selection, we compared the annual snow frequency using the  $2.5^{\circ}$ C threshold (alpine = 76.4%; subalpine = 61.5%) to a bivariate binary logistic regression phase prediction model (alpine = 76.7%; subalpine = 62.8%). This model predicts precipitation phase as a function of relative humidity and air temperature, and it was shown to the best precipitation phase method in a Northern Hemisphere comparison (Jennings et al., In Press). We have added this information to p. 6 (lines 10-16).

The temperature of precipitation is a likely shortcoming of SNOWPACK as the model sets precipitation temperature equal to air temperature. In independent work we have performed, we found wet bulb temperature to be a better predictor of new snow temperature, which was also noted in Harder and Pomeroy (2013). This should be included as an update in the SNOWPACK model considering wet bulb temperature could be easily estimated from the already standard forcing data. However, because SNOWPACK prescribes new snow temperature to be equal to air temperature, our estimates of the cold content added by precipitation are on the conservative side. The use of the colder wet bulb temperature (relative humidity is often below saturation, even during snowfall on Niwot Ridge) would lead to a greater amount of cold content added by precipitation. We have added this to the new modeling uncertainty discussion section (Sect. 5.2).

Fourth, despite reading through this a few times looking for it, it is unclear to me what kind of clearing this sub-alpine site is in. The site is specifically stated as a clearing, but the Snowpack canopy routine is enabled. This will significantly change the surface fluxes as well as precipitation at the snow surface; e.g., canopy interception. In my mind, this undermines the results presented herein – maybe it explains the poor result in Figure 3? – and needs to be detailed and the effects and impacts explained. Site photos would go a long way towards helping orient the reader. However as is, this is a major detail that is omitted.

This is an excellent point and we have added site photos to Figure 1. We have also changed the text from "small clearing" to "stand of lodgepole pine" to be clearer.

Fifth, A discussion on the role of Qg on cold content is needed and the assumptions behind your Qg simulation flux. These results show a treatment of the surface fluxes on cold content, but neglect discussion of soil-snowpack interactions, e.g., conditions that lead to frozen soil or refreezing of active layers.

We have added information to the manuscript discussing how  $Q_G$  is simulated (covered in detail in Reviewer 1's other  $Q_G$  comment on page 8 below). As to the rest of the comment, we do include  $Q_G$  in our analysis of the snowpack energy balance, noting that it is typically positive even during periods of cold content gain. Frozen soil is more relevant to runoff processes and is outside the scope of this work. Refreezing of active layers leads to cold content losses (latent heat is released as the phase of water changes from liquid to solid). While this process is important to snowpack ripening and snowmelt generation, we only consider the empirical relationships between cold content and snowmelt rate/timing in this work.

Lastly, the authors assert that increased peak cold content and total spring precipitation control snowmelt onset. But this seems by-definition – doesn't this imply more mass and refreshed albedos? Isn't this just what you'd expect with increased cold content being a function of snowpack mass?

This is noted in the discussion of the original manuscript:

"These results all suggest later seasonal snowmelt onset and faster snowmelt rates are primarily a function of persistent snowfall. While snowfall events can add significant cold content to the snowpack, they also change other fundamental properties that can delay snowmelt timing, such as increasing surface albedo (Clow et al., 2016) and adding dry pore space that must be saturated (Seligman et al., 2014)."

In summary: As I understand the results presented, the story is that the authors found limited evidence for sustained energy loss from the snowpack and that the cold content of the snowpack was mostly a result of mass inputs. However, there are many confounding factors that make it difficult to accept this at face

value. Given the circular reasoning in the results (more snow -> more cold content, but that is by definition), it is difficult for the reader to accept the results. That being said, validating the model against these observations is quite interesting and diagnosing snowpack energy loss during the winter is a useful contribution. However, I think the overall message needs to be refined to more clearly articulate the site-specific nature of this study, the uncertainties in key aspects of the analysis (e.g., precipitation, canopy), and the text improved for readability.

We again thank Reviewer 1 for their critical review of this work. We have added text to the manuscript to illustrate our results are specific to our two study sites in addition to the discussion section that covered this point in the original manuscript. We have also addressed their specific comments below to improve the readability of the text and more clearly outline the project's uncertainties.

Additionally, we would like to reiterate the novelty of this work. There is relatively little previous literature assessing the meteorological and energy balance controls on cold content development. We used a combination of observed data and validated simulation output to show precipitation was the dominant source of cold content development at our two sites. This finding was particularly surprising at the cold alpine site considering its high rates of snowpack sublimation and net longwave emission.

References Harder, P., and J. W. Pomeroy (2014), Hydrological model uncertainty due to precipitation-phase partitioning methods, Hydrol. Process., 28, 4311–4327, doi:10.1002/hyp.10214.

Specific points

Throughout:

The authors introduce (para 25) increase/decrease for cold content, but proceed to use gain/loss. I think it should be consistent throughout

We have changed p. 5 (lines 4-6) to include gain/loss and increase/decrease.

Figure is used in the text but Fig. when used in brackets. Ideally should be consistent.

This usage is in accordance with The Cryosphere's style guide (https://www.thecryosphere.net/for\_authors/manuscript\_preparation.html) and will be kept:

> "The abbreviation "Fig." should be used when it appears in running text and should be followed by a number unless it comes at the beginning of a sentence, e.g.: "The results are depicted in Fig. 5. Figure 9 reveals that..."."

Units should be separated with a cdot instead of spaces, e.g., Wâ'NE\*m(-2)

The Cryosphere does not specify the use of c-dots. The only example units we found in the manuscript prep instructions showed the units with a space as we have in the manuscript:

"Units must be written exponentially (e.g.  $W m^{-2}$ )."

Unclear what wet and dry days mean. Wet implies rain to me, but I suspect that's not what you mean. I would reword, or at least clearly define.

Yes, this is confusing. We have changed all relevant text to reflect this (snowfall/precipitation and non-snowfall/non-precipitation days instead of wet and dry days).

P1, Para 20: "cold content ... associated with reduced snowmelt" this needs to be reworded as snowmelt should be happening when CC = 0. Which melt rate is being considered?

Reworded for clarity.

P2, Para 20: "the authors" which authors?

Reworded for clarity.

P2, Para 25: "Furthermore: : :" I'm not sure I agree with this statement. CC needs to be = 0 for melt to occur, so isn't this known? Do you have a citation?

The citations are in the following paragraphs and they indicate they provide differing perspectives on the control exerted by cold content on seasonal melt rate and timing. However, our text was not clear enough (yes, melt occurs when  $Q_{net}$  is positive and CC = 0) that we were referring to winter cold content magnitude and we have edited p. 2 (line 29) for clarity.

P2, Para 30: "saturate", word choice

Saturate is commonly used in the snow hydrology literature in reference to satisfying the irreducible liquid water content of a snowpack, but we have edited p. 3 (lines 1-2) to be more specific and be applicable to other hydrologists to whom the word saturate may indicate all pore space is filled.

P2, Para 30: "However: ::", I'm unclear what you're trying to say, please clarify.

Edited for clarity.

P3, Para 10, 15, 30 Need to be indented.

Changed

P3, Para 20, use "10 m/s to 13 m/s" instead of how it is written.

Changed, but units are left in exponential form to be consistent with The Cryosphere style.

P4, Para 1, "Snow" incorrect capitalized

Snow begins an independent clause, meaning it can (or should, depending on your preferred style guide) be left capitalized.

P4, Para 14, "downwelling longwave" I would put a quick note as to what method you used.

Added this information to p.4 (lines 21-22), with full methodology detailed in the appendix.

P5, Para 20, remove "proposed in Sect. 1"

Changed

P5, Para 20 "We then quantified" I found this section unclear

Rewritten for clarity.

P5, Eqn 3 Consider writing 86,400 as a variable and showing in the text the units. Either way, you need units.

**Changed.**

P5, Para 15 "in order to improve" Using a model doesn't improve obs, it just compliments them. I think you should reword to make this distinction.

Yes, changed.

P5, Para 20 "number of finite elements" change to layers

Changed. Although SNOWPACK is a finite element model, that is not important here.

P5, Para 25 remove "the numerical model in"

Removed.

P5, Para 5, the canopy module stuff comes out of nowhere, especially given you say the site is in a clearing. This needs to be much clearer.

We changed the site description to note the pits are dug in a "stand" of lodgepole pine and have included a site photo in Figure 1.

P5, Para 20 "Output from snow model simulations" I don't follow. Do you mean the comparison is more robust w/multiple outputs to validate?

We are noting that the output of snow model simulations has greater fidelity when validated on more than just SWE, based on the work of Lapo et al. (2015). We are stating that we can make conclusions on the simulations of snowpack cold content because we have actual measurements of cold content to which we can compare the model output. We added text to p. 7 (lines 4-5) to clarify.

P6, Para 20 Any EC observations considered?

We did not consider using the EC observations on Niwot Ridge because the subalpine AmeriFlux tower measures fluxes above the canopy (21 m) and not near the snow surface. There are EC measurements in the alpine, but the records are short and the instruments are located near areas of snow scour.

P7, Eqn 4 The form for the energy balance equation given in Equation 4 is not a standard form. Generally, the change in internal energetics are given as a dU/dt and Qm is on the LHS. Qnet and Qm together are redundant in the energy balance as the energy available for melt is the net energy.

The form presented in the Equation 4 is used frequently throughout the snow hydrology literature (Cline, 1997; Marks et al., 2008; Marks and Dozier, 1992, to name just a few examples). However, we have changed the notation to the suggested form in order to avoid confusion. Additionally, we have changed  $Q_{net}$  to refer to the sum of the radiative, turbulent, and ground heat fluxes in order to avoid writing out the full energy balance each time we are referring to the net surface and ground fluxes.

P7, Para 1, "time scales" -> temporal scales

Time scale is appropriate and left as-is.

P7, Para 25, as I said above, I don't buy that an r2=0.17 demonstrates a primary control

Please see our previous note. We also clarified to say "of the two meteorological quantities evaluated here" to note we are referring to precipitation and air temperature. As mentioned previously, we devote an entire discussion section to the topic of subalpine cold content development and why precipitation exhibits a reduced effect relative to the alpine.

P8, Para 5, Probably should note these are depth averaged

Fixed.

P8, Para 10, -2.2 should have units after it P8,

Fixed.

Para 15, How is this working with the canopy module? Intercepted snow has massive sublimation losses, but that doesn't seem to be reflected here.

We are only concerned with snow surface sublimation as canopy sublimation does not directly lead to changes in snowpack internal energy.

P8, Para 20, Monotonically is either monotonic or not. There is no in-between. Reword

Correct. We removed that sentence and included a new sentence to clarify precipitation exerts a stronger control in the alpine than subalpine (p. 9, lines 17–21).

P8, Para 25, "simulations confirm" change to "support" or similar

**Changed.**

P9, Para 10, So how are you calculating Qg? Maybe I missed it? I think you need a reasonable treatment on the assumptions behind however you do this. Did you couple snowpack with the soil? Constant flux? Constant ground temp? Qg is important for a conduction heat flux into the snow pack, and needs to be addressed if you go after cold content. Often Qg is taken to be 0-4W/m2, but this flux can be important for stopping a numerical model from simulating absurd cold contents.

We used version 3.3 of SNOWPACK, which assumes a ground temperature of 0°C when there is snow cover. Simulations showed  $Q_G$  was typically between 0 and 4 W m-2 when snow depth exceeded ~20 cm, which is similar to other values reported in the literature. Thus,  $Q_G$  provides a small, but consistently positive to the snowpack energy balance. Cline (1997) noted ground heat flux was negligible at the alpine site using a flux plate in the soil. Snow pit data from the alpine and subalpine are consistent with this in that the warmest snowpack temperatures are observed at the bottom until ripening begins. We added this information to p. 6 (lines 20-21) in the methods and new Sect. 5.2 in the discussion.

P12, Para 10 "continued snowfall" But this is just more mass, so you'd expect snowmelt timing to be delayed P12,

Mass additions do not lead to consistent snowmelt responses because mass in and of itself is not a physical property that delays snowmelt (i.e., a given amount of warm, wet snow has a much different

effect on snow energetics and runoff than the same amount of cold, dry snow due to the amount of liquid water vs. dry pore space, changes to surface albedo, and cold content). Furthermore, we note other hypotheses in the discussion of the original manuscript:

"These results all suggest later seasonal snowmelt onset and faster snowmelt rates are primarily a function of persistent snowfall. While snowfall events can add significant cold content to the snowpack, they also change other fundamental properties that can delay snowmelt timing, such as increasing surface albedo (Clow et al., 2016) and adding dry pore space that must be saturated (Seligman et al., 2014)."

**Para 20, "future work: : :" Lots of work on this already: : :..**

We have rewritten to clarify that most previous work focuses on single, well-instrumented sites (such as our manuscript) or vast networks of SNOTEL-like sites with only air temperature, precipitation, and SWE observations (e.g., Trujillo and Molotch, 2014). A spatially explicit, energy balance treatment has yet to be applied to the different snow categories of the western United States despite the irreplaceable contribution of snowmelt to the hydrologic cycle and regional water resources. We have added text to this discussion section to clarify our statement (p. 14 lines 29-32).

P16, Para 30 Given Snowpack is forced hourly, this longwave estimate seems like a massive source of uncertainty, especially within the context of an energy balance model. There are many incoming longwave formulations that take into account various proxies for non-clear sky. You seem to do this for your emissivity, but it's not clear how that exactly works. With such low r2 this needs to be detailed and expanded upon. The large error in a critically important mid-winter energy flux may have substantial implications for this work.

This is an important point and we have added a note on this limitation to the new discussion section on modeling uncertainty (Sect. 5.2). Downwelling longwave radiation is under-sampled relative to other standard forcing data (Raleigh et al., 2016) and associated errors propagate into SWE and snow temperature biases (Lapo et al., 2015). Schlögl et al. (2016) showed little sensitivity in Alpine3D SWE simulations to a selection of empirical longwave estimates and Lapo et al. (2015) indicated the largest effects of longwave uncertainty were simulated at perturbations greater than  $\pm 10$  W m-2. Thus our small mean bias likely indicates the total amount of incoming longwave radiation is correct while the low r2 suggests the timing of subdaily fluctuations is not well simulated. Our multi-variable validation shows that SNOWPACK performs well relative to snow pit observations of SWE, depth-weighted snowpack temperature, and cold content.

Figures All figures – It would certainly aid readability to have them labeled as alpine/sub alpine without having to constantly refer to the caption.

Agreed. Changed on all relevant figures.

Figure 1, difficult to determine differences at high elevation.

We added contour lines to the figure along with site photos per an earlier recommendation.

Figure 2, can you change the DOY to dates for easier parsing?

Yes, changed for easier interpretation on this figure and others that previously showed DOWY.

Figure 5a,b Should have same axis extents

**Axes left as-is.**

Figure 8abcd would benefit from having the same y- (ab) and x- (cd) axes to aid in comparison. Also, please expand the y-axes of (ab) so-as to understand what the limits are.

Changed per this suggestion and our notes on the energy balance in pages 2 and 3 above.

Figure 9, needs legend

Changed.

The authors address the issue of the drivers of cold content evolution based on the exemple of two seasonal snowpacks from a single observation catchment in the Western US. They also assess, in a much shorter part, the effect of cold-content on snow-melt timing and rates.

The paper is well written and well illustrated. The take-home message is clear and the objectives assessed in Introduction are achieved with scientific quality. I found the paper both an appreciable synthesis of existing litterature on the topic, and enlightening regarding the conclusions achieved.

We thank Reviewer 2 for their suggestions and thoughtful review of the manuscript. Our responses are in blue text throughout this document and we have made changes to the manuscript in regard to their comments.

In addition to the few suggestions made below, there is in my opinion one minor contradiction in the Result section (point 9 below), that I would recommand the authors to adress with priority as it affects the consistency of the paper. Note that this apparent contradiction may come from misunderstanding from my side, or an edit mistake from the side of the authors. Comment addressed in point 9 below.

Introduction-p2 LI-6 : not the snowmelt itself is critical for the cited applications, but the timing of surface/subsurface runoff from Snowmelt, which may not be the same when surface melt occurs and refreezes (as just mentionned earlier in the manuscript). For consistence I suggest changing « snowmelt » into« runoff from snowmelt » in the current sentence.
 Yes, this is a good distinction. We have changed the text to reflect this (p. 2 lines 2-3).

 Introduction-p2 L16-18 : Aren't the « dominant processes »well-resolved by Method-3 (residual of the energy balance) at places where energy-balance models like SNOWPACK are routinely validated

against data regarding most components of the energy balance ? In that case, isn't it rather a lack of investigation into the process of cold content development, than a lack of validation data, that limits knowledge of the prevailing processes ?

Given the adequate performance of advanced energy balance models (Etchevers et al., 2004; Rutter et al., 2009), it is somewhat reasonable to assume they are accurately simulating the evolution of snowpack cold content. However, recent work has shown the utility in validating snow model output on more than one state variable (Lapo et al., 2015) in order to ensure we are not "getting the right answers for the wrong reasons." In this case we stand by our assertion that having measurements of snowpack temperature and cold content are necessary for making conclusions on how cold content develops in seasonal snowpacks. I.e., previous work could have tested hypotheses using energy balance model output alone, but a lack of cold content validation data would have limited the strength of their conclusions.

Furthermore, the dominant processes involved in cold content development likely depend on the climatology of the investigated sites. Event though it is well stated in the Discussion, specifying the perimeter of validity of your study should be done here already. Typically, I suggest transforming research question #1(p3 L6) into « What are the meteorological and energy balance controls on cold content development **at two alpine and sub-alpine sites from the Western US ?** »

Yes, we agree with this point and have changed research question 1 to reflect Reviewer 2's suggestion (p. 3 lines 9-10). This point was also noted by Reviewer 1, so we have added text wherever possible to note the results are specific to our study sites.

- 3. Introduction-p2 L22-26 : conduction fluxes within the snowpack, and from snowpack to the ground, can mitigate the impact of the intense negative fluxes reported here. If a gradient around 100 W/m develops within the snowpack as a result of intense surface cooling, around 20 W/m2 propagates downwards (upon hypothesis of a 0.2 W/m/K conductivity for Snow), which should somewhat prevents the snowpack from locally reaching unreaslitic temperatures (?) In the cited work, the authors note that their reported values are for ∆Q, or the change in snowpack internal energy, (Marks and Dozier, 1992). Although their paper has provided many significant contributions to the state of knowledge of the snowpack energy balance, it also underlines how little we previously knew about cold content development processes. We hope our manuscript provides a small step in the right direction.
- Introduction-p3 Ll : uncertainties->unknowns(suggestion) Changed.
- 5. Methods p5 L28 : « vapour diffusion » in SNOWPACK is actually only calculated to compute Snow grain/bounds growth rates. There is no mass redistribution between different snow layers as a result of vapour diffusion in current versions of SNOWPACK. I therefore suggest to suppress this item from the list of existing SNOWPACK routines, as it would be otherwise misleading. Thank you for clarifying. Given this paper is not about grain metamorphism, we have removed this part.

6. Methods- p6 L7-10 : could you specify here or in appendix the result of your calibration procedure for the parameters leaf area index, vegetation height, direct canopy throughfall, and wind speed reduction ? Note that these parameters are usually estimated from field data, and that any observation-based estimate of them would help assess the soundness of the calibrated parameter or of the canopy model. Yes. We have added this information to the Supplemental Material (Table S2). Additionnal, the rough size of the clearing where sub-alpine snowpits were made, should be specified (p4LI) to justify the use the canopy module of SNOWPACK, instead of an open- area version SNOWPACK with just wind attenuation.

We have included site photos for both locations (Figure 1) in the updated manuscript and removed the "small clearing" part from the text as this caused some confusion.

- Results-p5 L15-16 : « Peak cold content and peak SWE respectively occurred 33 d and 10 d later in the alpine than subalpine ».Add « on average » to this sentence and the next. Changed.
- 8. Results-p7L25-27 : « This is likely due to the increased variability of winter precipitation, the coefficient of variation of which is 2.9 and 2.7 times greater than that of air temperature in the alpine and subalpine, respectively ». I assume that by « increased »you mean« higher» ? I would suggest that snow-atmosphere heat transfers occuring during cold air temperatures periods are less efficient in cooling the

snowpack, than the direct addition of cold Snow from fresh snowfall.

Correct, we have changed *increased* to *higher*. Additionally, we have added text to Sect. 4.1 to note that the air temperature is less effective at producing cold content than precipitation.

9. Results- p7L29-30 : « During periods of SWE accumulation, Qnet was typically near 0 W m-2 (Fig. 4a), indicating a large negative energy balance was not responsible for cold content development. » First, here, you infer Qnet from the variation in CC between 2 snowpit dates, so where is the link to energy balance ? Second, based on Eq 3, Qnet ~0 W m-2 indicates no cold-content increase, meaning there is no visible snowfall-driven cold-content increase in the snowpit data. In my mind this contradicts the other results of the study, e.g. Fig 3 and 7 - please justify, or explain me where I am wrong. May I suggest using different names for Qnet in Eq. 3 and Qnet in Eq. 4 ? Like Qnet-pit and Qnet-Ea respectively. This is a good point and similar to the one Reviewer 1 brought up regarding our energy balance notation. In the quoted lines we were attempting to convey that the snow pits showed no direct evidence of a large negative surface energy balance like the one reported in Marks and Dozier (1992). We computed Qnet as a function of the change in cold content and the time between pit observations. To be more consistent, and clearer, we have changed our notation throughout the paper to have dU/dt be the change in internal energy of the snowpack (dU/dtpit for the snow pit data) and Qnet to represent the sum of SWnet, LWnet, QH, QLE, and QG (per the recommendation of Reviewer 1).

In regard to your point " $Q_{net} \sim 0 \text{ W m}^{-2}$  indicates no cold-content increase, meaning there is no visible snowfall-driven cold-content increase in the snowpit data", we have clarified in lines 20-27 (p. 8) the way we presented these data. Because cold content is a relatively small value in terms of W m-2, decreases in dU/dt will always be small, whether cold content gains come through precipitation or a negative surface energy balance. For example, if two snow pits were dug exactly one day apart, the computed dU/dt for a 0.2 MJ m-2 increase in cold content (2 cm of new SWE at -5°C) would be just -2.4 W m-2 (assuming all other energy balance components summed to zero). Thus, a fairly significant cold snowfall event would show up as a very small dU/dt value.

 Results- p8 L7 : could overestimated densities be the reason for cold-content overestimation at alpine location ? (as Snow temperature tend to be overestimated ?) Maybe a line on that could be added to the Result or Discussion section

We have added this point in the results section (p. 9 lines 3-6) and we have also added a new discussion section on the model shortcomings (Sect. 5.2).

**References**

[revised manuscript text omitted]

Keith Jennings 2/12/2018 3:26 PM Deleted: timing and rate Keith Jennings 2/12/2018 3:27 PM Deleted: are

Unknown Field Code Changed Keith Jennings 3/6/2018 4:2:

Keith Jennings 1/4/2018 9:47 AM **Deleted:** authors Keith Jennings 1/4/2018 9:47 AM **Deleted:** the Keith Jennings 3/7/2018 1:54 PM **Deleted:** net flux Keith Jennings 2/22/2018 12:47 PM **Deleted:** monthly energy required to satisfy cold content may be relatively small in comparison to the energy required to melt enough snow to fulfill the irreducible water content of an already isothermal snowpack (Bengtsson, 1982; Seligman et al., 2014).

[revised manuscript text omitted]

Keith Jennings 2/15/2018 11:57 AM Deleted: net energy flux Keith Jennings 2/15/2018 11:55 AM Keith Jennings 2/15/2018 11:56 AM Deleted: net Keith Jennings 2/15/2018 12:01 PM Deleted: 86.400 × Keith Jennings 2/15/2018 11:57 AM Deleted: Qnet Keith Jennings 2/15/2018 11:57 AM Deleted: i Keith Jennings 2/15/2018 11:58 AM Deleted: the net flux Keith Jenninas 2/15/2018 12:01 PM Deleted: 86,400 i Keith Jennings 2/15/2018 12:01 PM Formatted: Superscript Keith Jennings 2/13/2018 2:00 PM Deleted: improve on Keith Jennings 2/13/2018 2:01 PM Deleted: the temporal resolution of the snow pit observations, expand the study period, Keith Jennings 2/13/2018 2:01 PM Deleted: and Keith Jennings 2/15/2018 12:06 PM Deleted: It Keith Jennings 2/15/2018 12:07 PM Deleted: has also been Keith Jennings 2/15/2018 12:06 PM Deleted: previously

[revised manuscript text omitted]

Keith Jenning Deleted: s Keith Jennings 2/13/2018 2:17 PM Deleted: were Keith Jennings 2/13/2018 2:17 PM Deleted: e

Jennings 2/16/2018 9:25 Formatted: Font:Cambria Math, Italic Keith Jennings 2/22/2018 1:34 PM Formatted: Superscript Keith Jennings 2/16/2018 9:25 AM Formatted: Superscript

| 1 | Keith Jennings 2/15/2018 11:53 AM                        |
|---|----------------------------------------------------------|
|   | Deleted: Q net                                |
|   | Keith Jennings 2/15/2018 12:28 PM                        |
|   | Deleted: $-Q_M$                                   |
| 1 | Keith Jennings 2/15/2018 12:29 PM                        |
|   | Formatted: Superscript                                   |
| 1 | Keith Jennings 2/15/2018 12:29 PM                        |
|   | Deleted: ,                                               |
|   | Keith Jennings 2/15/2018 12:29 PM                        |
| J | Deleted: , and                                           |
|   | Keith Jennings 2/15/2018 12:28 PM                        |
|   | Deleted: $Q_{M}$ is the energy available for melt |

Keith Jennings 2/22/2018 2:10 PM Deleted: sub-Keith Jennings 2/22/2018 2:10 PM Deleted: sub-

**4 Results**

**4.1 Snow pit observations of cold content**

Snow pit observations showed daily and peak annual snowpack cold content were consistently greater in the alpine than subalpine (Fig. 2). From WY2007–WY2013, mean peak cold content was 2.6 times greater in the alpine than subalpine, while mean peak SWE was 2.1 times greater in the alpine (Table 1). On average, peak cold content and peak SWE respectively occurred 33 d and 10 d later in the alpine than subalpine. The average temporal gap between peak cold content and peak SWE was also 23 d shorter in the alpine, indicating greater energy exchange between the snow and atmosphere at this site during the main time of snowpack ripening. Mean  $\frac{dU}{dt_{pit}}$  for this period, as estimated using Eq. 3, was 1.2 W m-2 and

 $0.4 \text{ W m}^{-2}$  in the alpine and subalpine, respectively.

10

5

From 1 December to the date of snow pit observation, increased cumulative precipitation was associated with increased cold content at both sites (Fig. 3). Cumulative precipitation explained 55% and 17% of the variance in cold content in the alpine and subalpine, respectively. The relationship was statistically significant at the 99% level at both sites despite the low coefficient of determination in the subalpine. Conversely, the cumulative mean of air temperature had no statistically significant relationship to snowpack cold content, explaining less than 1% of the variance at both sites (not shown).

- 15 Although there may be snowpack energy losses during periods of cold air temperature, these results indicate that, of the two meteorological quantities evaluated here, snowfall exerts the primary control on cold content development. This is likely due to the higher variability of winter precipitation, the coefficient of variation of which is 2.9 and 2.7 times greater than that of air temperature in the alpine and subalpine, respectively. Furthermore, the difference in  $r_{\rm a}^2$  values between the two sites suggests that precipitation plays a more important role in the alpine than subalpine in terms of cold content development.
- 20

Snow pit observations were also used to  $\underbrace{calculate}_{t} \frac{dU}{dt_{plt}}_{plt}$  by quantifying the change in cold content between two points in time (Eq. 3). During periods of SWE accumulation,  $\frac{dU}{dt_{plt}}_{plt}$  was typically near 0.0 W m-2 (Fig. 4a), indicating a large negative energy balance was not responsible for cold content development at our two sites. The average flux in the alpine (-0.8 W m-2) was greater in magnitude during this period than in the subalpine (-0.4 W m-2), and both distributions were left-skewed as the energy balance was typically negative from snowfall-and/or flux-driven cold content increases. Changing the

25 analysis to snow pit observations when melt occurred (Fig. 4b) led to a pronounced right-skew in the flux distribution with values again of a higher magnitude in the alpine. Thus, we found no evidence for highly negative internal energy changes at our sites with  $\frac{dU}{dt_{pit}}$  values only being large in magnitude during snowmelt.

**4.2 Model SWE, snowpack temperature, and cold content validation**

SNOWPACK simulations reproduced observed snow pit SWE patterns at both sites, with a higher coefficient of determination and lower bias in the subalpine than alpine (Fig. 5a,b; Table 2). Subalpine simulations were also in line with

8

**Keith Jennings 2/13/2018 10:34 AM Deleted: Pe Keith Jennings 2/13/2018 10:34 AM Deleted: Keith Jennings 2/16/2018 9:45 AM Deleted: Qint**

Keith Jennings 2/16/2018 9:50 AM Deleted: with Keith Jennings 3/7/2018 3:30 PM Deleted: T Keith Jennings 2/13/2018 10:24 AM Deleted: increased Keith Jennings 2/16/2018 9:46 AM Formatted: Superscript Keith Jennings 2/15/2018 12:34 PM Deleted: infer Keith Jennings 2/15/2018 12:34 PM Deleted: Qnet Keith Jenning 2/15/2018 12:35 PM Deleted: Qnet Keith Jennings 3/6/2018 4:53 PM Formatted: Superscript Keith Jennings 3/6/2018 4:53 PM Formatted: Superscript

[revised manuscript text omitted]

**Keith Jennings 2/21/2018 4:07 PM**

| Keith Jennings 3/7/2018 2:16 PM    |  |
|------------------------------------|--|
| Deleted: is                        |  |
| Keith Jennings 3/7/2018 2:16 PM    |  |
| Deleted: to                        |  |
| Keith Jennings 3/7/2018 2:16 PM    |  |
| Deleted: in snow model simulations |  |

**Keith Jennings 2/14/2018 8:34 AM**

**Comment [1]:** Reword to indicate that method 3 is important by first prinicples but that 2 is more efficient, at least at the sites studied here

Keith Jennings 2/22/2018 2:18 PM Deleted: Our results also indicate method 3

provides utility in simulating cold content development, but to a lower degree than method 2. Additionally, w

**Keith Jennings 2/22/2018 2:19 PM Deleted: Periods Keith Jennings 3/7/2018 2:14 PM Deleted: short, Keith Jennings 2/22/2018 2:19 PM**

Keith Jennings 2/22/2018 2:18 PM

widely validated, physics-based SNOWPACK model. Despite our adherence to such protocols, there are still significant sources of uncertainty inherent to model-based snow studies.

Snow model intercomparison work has consistently shown there is no one best model and that model performance

- varies between and within sites and water years (e.g., Boone and Etchevers, 2001; Essery et al., 2013; Etchevers et al., 2004;
   Rutter et al., 2009; Slater et al., 2001). This body of research acknowledges that all models imperfectly represent snow cover evolution and the snowpack energy balance. One example shortcoming of SNOWPACK relevant to the work presented herein is that the temperature of new snow is set to be equal to air temperature despite the fact that hydrometeor temperature is more accurately estimated as a function of the psychrometric energy balance (e.g., Harder and Pomeroy, 2013). Using the psychrometric approach gives snowfall a temperature near the wet bulb temperature, which is lower than air temperature to when relative humidity is under 100% (Harder and Pomeroy, 2013). Thus, the temperature of new snow is likely to be
- overestimated by SNOWPACK, while cold content additions are underestimated. This means our computation of the total cold content contributed by precipitation is likely on the conservative side as using the wet bulb temperature would lead to increased cold content gains during snowfall.
- Another source of uncertainty in our work is the use of an empirical method to estimate incoming longwave radiation as a function of air temperature, relative humidity, and incoming shortwave radiation (Appendix A). Recent research has shown errors in incoming longwave radiation propagate into SWE, snow surface temperature, and energy balance biases (Lapo et al., 2015; Raleigh et al., 2016). We aimed to reduce the error in our incoming longwave radiation estimates by using the recommended clear sky and cloud correction protocols for Niwot Ridge (Flerchinger et al., 2009). At both the alpine and subalpine site, the mean biases were within the instrument range of error when compared to shorter-term observations,
- 20 indicating the total estimated amount of incoming longwave radiation was acceptable. However, the low  $r_{k}^{2}$  of the hourly estimates suggests the sub-daily fluctuations of incoming longwave radiation were not well simulated. Despite these issues, model performance was high in terms of simulated SWE, depth-weighted snowpack temperature, and cold content (Sect. 4.2). This may due to compensatory errors in the model (Etchevers et al., 2004; Kirchner, 2006) or because SNOWPACK is relatively insensitive to the choice of incoming longwave radiation estimate (Schlögl et al., 2016).
- 25 Additionally, we had no long-term ground surface temperature data to force the model, so we used the SNOWPACK default value of 0°C. This produced mean  $Q_G$  values of 2.0 W mx-2 and 0.8 W mx-2 during periods of SWE > 1 cm in the alpine and subalpine, respectively. Previous work from Niwot using a heat flux plate indicated  $Q_G$  in the alpine to be negligible (Cline, 1997), while other researchers showed the upper layer of alpine soil could approach temperatures significantly below freezing during periods of shallow snow cover (Brooks and Williams, 1999). Therefore, the SNOWPACK-simulated alpine
- 30  $Q_G$  is likely an overestimate. In the subalpine, the soil temperature at 5 cm below the surface is typically between -1°C and 0°C during the winter (Burns et al., 2014), meaning the use of the default 0°C ground surface temperature is reasonably in agreement with shorter term observations.

Keith Jennings 2/19/2018 11:43 AM Formatted: Indent: First line: 0.31"

Keith Jennings 2/19/2018 11:46 AM Formatted: Superscript

Keith Jennings 3/7/2018 2:31 PM Formatted: Superscript Keith Jennings 3/7/2018 2:31 PM Formatted: Superscript

Keith Jennings 2/19/2018 11:42 AM Formatted: English (US)

**5.3 Differences between cold content development controls in the alpine and subalpine**

Despite only a 506 m elevation difference between the two sites, the role of a negative energy balance in developing cold content in the subalpine was approximately double that of the alpine. Simulations of snowpack temperature indicated the increased sensitivity was likely due to the shallower subalpine snow depth. Diurnal snowpack temperature range generally

- decreases with depth (e.g., Burns et al., 2014; DeWalle and Rango, 2008; Sturm et al., 1995) and our simulations showed daily fluctuations to be largest in the snowpack's upper layers, converging towards 1.0°C as depth exceeded 500 mm (Fig. 13). This is the same depth at which the insulating effects of snow on soil temperature become marginal (Slater et al., 2017). Likely this is because the penetration of incoming shortwave radiation and sensible heat transfer through windpumping are limited to the top portion of the snowpack (Albert and McGilvary, 1992; Colbeck, 1989a, 1989b; Lehning et al., 2002b),
- 10 while the low thermal conductivity of snow modulates energy transfer below the active upper layers (Sturm et al., 1997). In this case, proportionally more of the shallower subalpine snowpack was interacting with surface energy exchange, making it more sensitive to positive and negative fluxes. Furthermore, subalpine cold content was consistently lower in magnitude, meaning it took less energy input to drive cold content to zero and relative fluctuations were larger. Therefore, shallower snowpacks with reduced cold content, like those in the subalpine, are more susceptible to relatively rapid changes in internal
- 15 energy from surface energy fluxes.

**5.4 Other controls on seasonal snowmelt timing and rate**

Previous research has suggested uncertainty in the degree to which cold content controls snowmelt timing at daily to seasonal time scales. In our research, we found no statistically significant relationship between peak cold content magnitude and seasonal snowmelt onset using data from both observations and simulations. Rather, the majority of the variance in

- 20 seasonal snowmelt onset was explained by the timing of annual peak cold content and total spring precipitation. Later peak cold content generally occurred due to cold spring storms depositing significant snowfall. If such events were then followed by continued snowfall, then snowmelt timing was delayed. Meanwhile, seasonal snowmelt rate, or the ablation slope, was primarily controlled by peak SWE magnitude and timing, with greater, later peak SWE corresponding to more rapid snowmelt.
- 25 These results all suggest later seasonal snowmelt onset and faster snowmelt rates are primarily a function of persistent snowfall. While snowfall events can add significant cold content to the snowpack, they also change other fundamental properties that can delay snowmelt timing, such as increasing surface albedo (Clow et al., 2016) and adding dry pore space that must be saturated (Seligman et al., 2014). Other research shows seasonal snowmelt onset is also related to air temperature (Kapnick and Hall, 2012) and snow surface impurities (Painter et al., 2010; Skiles et al., 2012). Although much

30

work has been done evaluating the empirical controls exerted by snowpack and climatic properties on snowmelt rate and timing across large spatial extents (e.g., Trujillo and Molotch, 2014), relatively little research has been done at such scales on the physical processes (e.g., cold content and the snowpack energy balance). Given the importance of seasonal snowmelt

14

Keith Jennings 2/19/2018 11:42 AM

Keith Jennings 3/7/2018 2:44 PM Deleted: 0.3°C to 0.5

**Keith Jennings 2/19/2018 11:42 AM Deleted: 3**

Keith Jennings 2/19/2018 10:35 AM Deleted: Additionally, Keith Jennings 2/19/2018 10:35 AM Deleted: o Keith Jennings 2/19/2018 10:32 AM Deleted: suggests Keith Jennings 2/22/2018 2:24 PM Deleted: affected Keith Jennings 2/22/2018 2:24 PM Deleted: by timing to water resources management and various hydrologic processes, future synthesis work should evaluate the effect of various physical processes on snowmelt rate and timing across snow-dominated regions globally, leveraging both field observations and physics-based snow model simulations.

**5.5 Cold content development processes in other seasonal snow classes and climates**

- 5 Despite the research presented here, there are still unanswered questions regarding cold content development as well as its effect on snowmelt rate and timing. Firstly, we have only presented results from two sites within a single snow-dominated research catchment. Seasonal snow cover in the western United States spans a large elevational gradient and includes both maritime (e.g., the Cascades and Sierra Nevada) and continental (e.g., the Rocky Mountains) snowpack regimes (Armstrong and Armstrong, 1987; Serreze et al., 1999), Globally, seasonal snow cover includes an even greater number of classes,
- 10 including the cold, thin snowpacks of the Arctic and the Canadian Prairies (Sturm et al., 1995). Therefore, an avenue for future research is to examine differences in cold content development across seasonally snow covered areas, with a particular focus on disentangling the effects of precipitation and air temperature during snowfall at sites with different snowpack characteristics. For example, snowpacks in California's Sierra Nevada are typically deep, but air temperature, is generally near freezing, even during winter storm events. Considering the cold content of precipitation is a linear function of air
- 15 temperature and precipitation depth (Eq. 2), a given unit of snowfall in the Sierra Nevada should contribute less snowpack cold content than that same unit in the colder Rocky Mountains. Therefore, the control that precipitation exerts on cold content development is likely different between the two locations. Additionally, it is uncertain how our results translate to cold, shallow tundra and taiga snowpacks. In this study, we observed marked differences in cold content development processes between the alpine and subalpine, with the energy balance exerting greater control in the shallower subalpine snowpack. It may be that the energy balance is of even greater importance in tundra and taiga snowpacks, but further work is
- needed.

Secondly, a large amount of recent literature has shown unequivocally that, due to climate warming, patterns of snow accumulation and melt are changing across the globe with resultant effects on myriad hydrologic processes (Barnhart et al., 2016; Berghuijs et al., 2014; Knowles et al., 2006; Mote et al., 2005; Musselman et al., 2017; Pederson et al., 2011; Stewart,

- 25 2009). It is uncertain what role, if any, cold content plays in the climate-driven changes on snow processes. In our investigations we found pit-observed SWE was a strong predictor of cold content (alpine  $r^2 = 0.84$ ; subalpine  $r^2 = 0.50$ ), with subalpine cold content lower per unit SWE due to warmer depth-weighted snowpack temperatures. Both sites also exhibited a significant positive linear relationship between the cumulative mean of air temperature and snowpack temperature. Therefore, a unit of SWE in a warmer location or climate should correspond to reduced cold content due to increased
- 30 snowpack temperature. Our work showed that decreased cold content magnitudes corresponded to faster snowmelt rates and earlier snowmelt timing at time scales less than 1 month. Therefore, reductions in snowpack cold content due to climate warming have implications for meltwater timing and availability, which could impact water resources management.

15

Keith Jennings 2/19/2018 10:42 AM **Deleted:** focus on disentangling

Keith Jennings 2/19/2018 11:42 AM Deleted: 4

**Unknown Field Code Changed Keith Jennings 2/14/2018 8:35 AM Deleted: (Serreze et al., 1999; Sturm et al., 1995) Keith Jennings 2/21/2018 4:17 PM Deleted: further**

Keith Jennings 2/22/2018 2:26 PM Deleted: s Keith Jennings 2/22/2018 2:26 PM Deleted: are Keith Jennings 2/14/2018 8:54 AM Deleted: Given Keith Jennings 2/14/2018 8:49 AM

**6** Conclusions**

We have presented an analysis of snowpack cold content using data from a long-term snow pit record and 23 y of physicsbased snow model simulations at an alpine and subalpine site within the Niwot Ridge LTER. The research questions were designed to fill important missing gaps in the snow hydrology literature, namely the meteorological and energy balance

- 5 processes behind cold content development and how cold content controls snowmelt rate and timing Observations and simulations showed new snowfall was the primary pathway for cold content development at our sites, being responsible for 84.4% and 73.0% of modeled daily cold content gains in the alpine and subalpine, respectively. Snowfall days with cold content gains outnumbered non-snowfall days with gains by a 4.2:1 ratio in the alpine and 2.6:1 in the subalpine. A negative energy balance—averaging > - $\frac{2}{3}$ 0 W m-2 in the alpine and subalpine—was responsible for the remainder of cold content
- 10 gains, primarily due to the cooling effect of sublimation and net longwave emissions. At subdaily time scales, dry-period cold content increases occurred preferentially at night at both sites. We found no evidence in either the snow pit record or the simulation data for large negative energy fluxes generating significant snowpack cold content. Additionally, air temperature showed little to no relationship to cold content development at either of the sites we studied.

Seasonal snowmelt timing was not significantly correlated with peak cold content magnitude, but rather the timing of peak cold content and total spring precipitation controlled snowmelt onset. Later peak cold content and increased spring precipitation delayed snowmelt in both the alpine and subalpine, explaining 84.7% and 61.4% of the variance in peak SWE timing. Cold content magnitude did affect sub-seasonal snowmelt in that non-zero initial cold content values corresponded to delayed snowmelt timing and slower snowmelt rates. At daily time scales, the majority of melt events and the fastest melt

- rates occurred only when  $CC_{6AM} = 0.0 \text{ MJ m}^{-2}$ . Any existing energy deficit at 6AM damped daily snowmelt rates.
- 20 The Niwot Ridge LTER provided the ideal study location for the research presented in this paper. The site's unique long-term snow pit and hourly meteorological records facilitated in-depth analyses into snowpack processes using both observations and physics-based snow model simulations. Lacking either data source would have limited the scope of this paper and added further uncertainty. Therefore, we hope this work underlines the utility of long-term *in situ* snowpack and meteorological measurements as they allow for robust analyses on the observations themselves and also enable model
- 25 validation on multiple snowpack properties (e.g., mass, depth-weighted temperature, and cold content), which improves the quality of simulated output.

**Data availability**

30

The quality controlled, infilled meteorological dataset presented in this work is hosted on the Niwot Ridge LTER website (http://niwot.colorado.edu/data/data/infilled-climate-data-for-cl-saddle-dl-1990-2013-hourly). Please use this paper as the data citation and contact KSJ with questions (Keith.Jennings@colorado.edu). Snow pit

(http://niwot.colorado.edu/index.php/data/data/snow-cover-profile-data-for-niwot-ridge-and-green-lakes-valley-1993-ongoi) and precipitation data (http://niwot.colorado.edu/index.php/data/data/precipitation-data-for-c1-chart-recorder-1952-ongoing

16

**Keith Jennings 3/6/2018 8:01 PM Deleted: long-term Keith Jennings 2/22/2018 2:30 PM Formatted: Indent: First line: 0"**

**Keith Jennings 2/22/2018 2:30 PM**

**Deleted:** . To improve on the temporal resolution of the snow pit record, we ran the physics-based SNOWPACK model with a quality controlled, serially complete hourly dataset from WY1991– WY2013, a period covering a wide range of snowpack conditions.

Keith Jennings 3/7/2018 2:47 PM

Keith Jennings 3/7/2018 2:47 PM Deleted: dry

Keith Jennings 2/22/2018 2:30 PM Deleted: 6

Keith Jennings 3/7/2018 2:49 PM Deleted: a high level of analysis

Keith Jennings 3/7/2018 2:49 PM Deleted: in-depth

Keith Jennings 2/22/2018 2:31 PM Deleted: will be posted Keith Jennings 2/22/2018 2:31 PM Deleted: http://niwot.colorado.edu/index.php/data/ 
[revised manuscript text omitted]

28